# THEORY-INSPIRED TASK-RELEVANT REPRESENTATION LEARNING FOR INCOMPLETE MULTI-VIEW MULTI-LABEL LEARNING

## ABSTRACT

Multi-view multi-label learning is commonly hindered by dual data incompleteness, arising from constraints in feature collection and prohibitive annotation costs. To address the intricate yet highly practical challenges and enhance the reliability of representation extraction, heterogeneous feature fusion, and label semantic learning, we propose a Theory-Inspired Task-Relevant Representation Learning method named TITRL. From an information-theoretic standpoint, we identify the sources of view-specific information that interfere with shared representations. By introducing dual-layer constraints on feature exclusivity and label integration, TITRL constructs a general framework for task-relevant information extraction. Besides, through variational derivation, we demonstrate the existence of tractable bounds for the mutual information model that guides the optimization direction. Regarding label semantic learning, we establish flexible relationships between label prototypes by promoting the expression of sample-level label correlations. During the multi-view integration process , TITRL simultaneously incorporates early fusion through distribution information aggregation and late fusion weighted by prediction confidence, which improves the semantic stability while enabling dynamic view quality assessment. Finally, extensive experimental results validate the effectiveness of TITRL against state-of-the-art methods.

## 1 INTRODUCTION

Integrating information from diverse sources enables richer insights and a more holistic understanding of complex problems. By capitalizing on the rapid proliferation of multimodal data, multi-view learning shows strong potential to deliver superior performance across numerous application domains Wen et al. (2022); Fang et al. (2021; 2022). At the same time, multi-label classification, where one instance may correspond to multiple relevant labels, has become increasingly important given the growing annotation demand in the information era. For instance, in remote sensing, an image may be simultaneously tagged with labels such as "forest", "urban area", and "water body"Sumbul et al. (2019). Multi-view features provide comprehensive representations of objects, and multiple labels capture their diverse attributes. These characteristics address the limitations of single-view and single-label paradigms in traditional machine learning, aligning with the demands of real-world applicationsQin et al. (2025); Tian et al. (2024). By integrating multi-view learning with multi-label classification, a thorough instance depiction and the enriched information for the recognition of multiple labels are attained. Therefore, multi-view multi-label classification (MvMLC) has emerged as a highly promising avenue of research Liu et al. (2025); Zhang et al. (2018).

Existing MvMLC techniques seek to leverage heterogeneous features and predict multiple labels within a unified framework. Representative methods include lrMMC Liu et al. (2015) applying low-rank matrix factorization, and $\text{E}F^2\text{FS}$ Hao et al. (2025) based on feature selection. However, many approaches still rely on the assumption that both complete views and full label sets are accessible, which rarely holds in practice. In reality, multi-view data often suffer from missing modalities owing to feature acquisition and processing difficulties. For example, in remote sensing, multispectral imagery may be collected while hyperspectral or LiDAR data are absent due to sensor limitations or high storage requirements Guan et al. (2025). Similarly, multiple annotations are frequently incomplete since labeling cost is expensive, privacy restrictions prevent data sharing, or some categories remain semantically ambiguous. In medical imaging, chest X-rays may contain multiple pathologies but only a subset is annotated, as labeling requires domain expertise and clear diagnostic boundaries are sometimes lacking Sun et al. (2024). The presence of numerous features and labels, coupled

with concurrent data missingness constitutes a widespread challenge, making incomplete multi-view multi-label classification (iMvMLC) particularly complex and urgent to address.

With the advancement of deep learning, various methods based on different network architectures have been applied to address the iMvMLC problem. Nevertheless, these methods still present opportunities for refinement, especially with regard to feature representation extraction, view fusion, and the construction of label semantics. (i) Enhancing information sharing across multiple views is a pivotal factor in both unsupervised clustering tasks Zhou et al. (2024) and supervised classification tasks Chen et al. (2024). DICNet Liu et al. (2023b) and LMVCAT Liu et al. (2023c) capture shared representations by utilizing cross-view interaction mechanisms. However, these methods fail to account for the disruptions caused by view-specific information. As a result, redundancy and noise are inadvertently incorporated into the shared representations, diminishing their purity and increasing the risk of misguiding the classification process. Although SIP Liu et al. (2024b) is a method for minimizing non-shared information and maintaining feature validity, it does not integrate label information to guide representation extraction, which leads to uncertainty about the practicality of the obtained common information. (ii) Previous approaches, such as DIMC Wen et al. (2023) and AIMNet Liu et al. (2024a), Wen et al. (2023); Liu et al. (2024a) have largely focused on feature-level weighting for view fusion. Nevertheless, without leveraging classification confidence as a weighting signal, such methods often fail to capture discriminative information from each view. As the number of categories increases, it becomes crucial to identify pertinent information for predicting each category, which underscores the need for label-specific feature selection. (iii) Learning multi-label semantics necessitates modeling label relationships. Methods like MTD Liu et al. (2023a), which treat multi-label learning as separate binary classifications, are inherently limited in achieving optimal performance. Moreover, label correlations cannot be regarded as fixed pairwise measures, as assumed in traditional methods. The realization of label correlations often fluctuates between different instances Si et al. (2023). For example, in a movie recommendation system Li et al. (2025), the correlation between "action" and "adventure" genres may be stronger for some users, while weaker for others, depending on individual preferences.

To address these problems, we propose a Theory-Inspired Task-Relevant Representation Learning framework named TITRL. The motivation behind TITRL is to enhance the purity of shared representations, improve the effectiveness of view fusion, and delicately capture the multi-label correlation semantics. We begin by leveraging mutual information-based semantic interaction and theoretically establishing a dual-layer constraint framework at the levels of feature and category. Guided by the principle of mitigating view-specific noise that adversely affects representation extraction and downstream prediction, we disentangle the view-specific mixtures that indicate the negative influence of each view on label recognition. Besides, we obtain tractable bounds for the mutual information model through variational derivation, which serves as the training loss to guide the extraction of common information. Regarding view fusion, we initially employ a distribution-aware blending strategy to derive the distribution parameters of the integrated shared information, which not only aids in selecting views with stable statistical properties but also facilitates coherent posterior distribution inference. After constructing the prototype representation for each label, the pseudo-labels are generated by leveraging the interactions between view representations and these prototypes. Subsequently, we perform a confidence-based late fusion by utilizing the pseudo-labels derived from the remaining views after removing each individual view, along with the prediction from all views. The process aims to mitigate the view-specific interference while retaining the most informative insights that contribute to label prediction. Finally, to accurately model label correlations, we focus on maximizing the similarity between the positive label prototypes of each sample and its shared representation. This approach promotes the learning of a sample-specific correlation structure, which enables flexible utilization of label dependencies to improve classification performance. The main contributions of our TITRL are summarized as follows:

- We propose a general framework for multi-view shared representation extraction, applying constraints at both the feature and label levels. Moreover, we theoretically establish the optimization direction of the model and derive the variational bound to guide the training process.

- TITRL simultaneously considers the statistical properties of representation extraction and the confidence of label prediction in view fusion. Additionally, TITRL proposes a flexible approach to represent label correlations, which focuses on the diverse manifestation patterns across samples.

- Extensive experimental results across a range of public datasets and varying degrees of data missingness demonstrate the effectiveness and robustness of our method.

## 2 METHOD

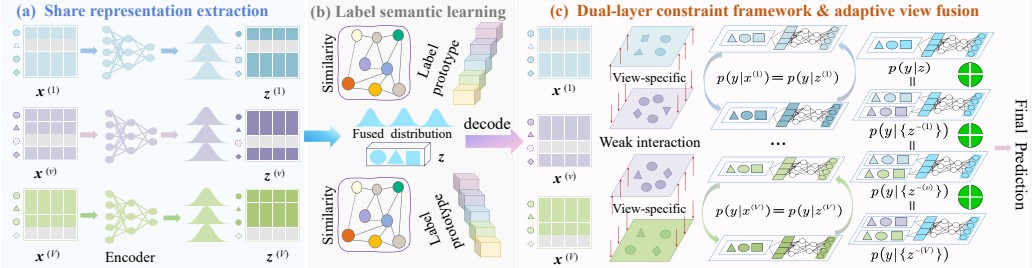

Figure 1: The main framework of our proposed TITRL. Different shapes signify different samples.

### 2.1 PROBLEM DEFINITION

Consider a dataset consisting of $n$ labeled instances, represented as $(\{\boldsymbol{x}^{(v)}\}_{v=1}^{V}, \boldsymbol{y})$, where each sample is observed from $V$ distinct views. Specifically, the $v$-th view of any sample is denoted as $\boldsymbol{x}^{(v)} \in \mathbb{R}^{d_v}$, while the associated label $\boldsymbol{y} \in \{0,1\}^c$ corresponds to $c$ categories. Additionally, we define $\mathcal{V}$ ($|\mathcal{V}| \leq V$) as the set of observed views. Thus, the available multi-view data can be expressed as $\{\boldsymbol{x}^{(v)}\}_{v \in \mathcal{V}}$ (abbreviated as $\{\boldsymbol{x}\}$). Moreover, let $\mathcal{U}$ represent the set of known tags, where $|\mathcal{U}| \leq c$. The goal is to design an end-to-end neural network capable of performing classification tasks on incomplete multi-view partial multi-label data.

### 2.2 TASK-RELEVANT REPRESENTATION LEARNING UNDER A DUAL-LAYER CONSTRAINT FRAMEWORK

Enhancing cross-view information interaction in multi-view learning has consistently been a crucial driver of improved classification performance. Moreover, prior researches Federici et al. (2020) have demonstrated that integrating the common information across all views and reducing the redundant information introduced by view-specific factors is sufficient to accomplish all prediction tasks. For example, in facial recognition Liu et al. (2023d), images from different views capture shared facial features, with the frontal view revealing details of the eyes and nose, and the side view presenting the contours. However, some views may introduce disruptive factors, including lighting variations, background clutter, or excessive emphasis on minor details, which can disrupt model performance. By integrating shared features and removing noisy information, the model can achieve more accurate recognition. Given an initial shared representation $\boldsymbol{z}^{(v)} \in \mathbb{R}^d$ for each view, the unified representation $\boldsymbol{z} \in \mathbb{R}^d$ is obtained by fusing them. To guarantee that the shared representation captures the common information across all views, it is crucial for the semantics of $\boldsymbol{z}$ to encompass the relevant information from original views as much as possible. This objective introduces the requirement of optimizing the mutual information interactions between $\boldsymbol{z}$ and each individual view to their fullest extent, i.e., $\max \frac{1}{|\mathcal{V}|} \sum_{v \in \mathcal{V}} I(\boldsymbol{x}^{(v)}; \boldsymbol{z})$. Moreover, the detrimental redundancy arising from the distinct information inherent to each view should be meticulously minimized, which necessitates that the derived representation primarily conveys the shared components, while effectively attenuating the noise caused by view-specific characteristics to the absolute minimum. Thus, the representation $\boldsymbol{z}^{(v)}$ ought to be distanced from the view-specific information from other perspectives, with the aim of minimizing $I(\{\boldsymbol{x}^{\sim(v)}\}; \boldsymbol{z}^{(v)}|\boldsymbol{x}^{(v)})$, where $\{\{\boldsymbol{x}^{\sim(v)}\}, \boldsymbol{x}^{(v)}\} = \{\boldsymbol{x}\}$. Next, due to the scalability of information transfer, we can derive the following upper bound to guide information separation:

$$\sum_{v \in \mathcal{V}} I(\{\boldsymbol{x}^{\sim(v)}\}; \boldsymbol{z}^{(v)}|\boldsymbol{x}^{(v)}) \leq \sum_{v \in \mathcal{V}} I(\{\boldsymbol{x}^{\sim(v)}\}; \boldsymbol{z}|\boldsymbol{x}^{(v)}). \tag{1}$$

We have concentrated on the suppression of view-specific redundancy at the feature level. However, it remains uncertain whether these representations are directly applicable to downstream classification as label information is not integrated. Therefore, it is crucial to incorporate task-specific knowledge to steer the unification of these features toward enhancing classification performance. Foremost, it is imperative to prevent information degradation by ensuring that the extracted representations preserve the mutual information between the original features and their corresponding labels. This requirement imposes the exact equivalence between $I(\boldsymbol{x}^{(v)}; \boldsymbol{y})$ and $I(\boldsymbol{z}^{(v)}; \boldsymbol{y})$:

$$\min \sum_{v \in \mathcal{V}} (I(\boldsymbol{x}^{(v)}; \boldsymbol{y}) - I(\boldsymbol{z}^{(v)}; \boldsymbol{y})). \tag{2}$$

In addition, another crucial consideration lies in ensuring the extracted information is solely label-relevant and devoid of any admixed noise. In this regard, by isolating the distinctive impact of $z^{(v)}$ within the task-relevant components, we obtain the following expression:

$$I(\boldsymbol{y}; \boldsymbol{z}^{(v)}) = \underbrace{\sum_{j=1, j \neq v}^{V} I(\boldsymbol{y}; \{\boldsymbol{z}^{\sim(j)}\}|\boldsymbol{z}^{(j)}) + I(\boldsymbol{y}; \{\boldsymbol{z}\})}_{\text{shared } I_v^s} + \underbrace{I(\boldsymbol{y}; \boldsymbol{z}^{(v)}|\{\boldsymbol{z}^{\sim(v)}\})}_{\text{view-specific}}, \qquad (3)$$

where the preceding term is referred to as shared information, as each of its components encapsulates associative information contributed collectively by multiple views toward the label. Then, our optimization goal is to achieve cleaner feature extraction by controlling task-irrelevant information, reduce misclassifications caused by view-specific redundancy, and emphasize the collaborative discriminative power of all useful signals from multi-view. Since the shared information term $I_v^s$ consists of multiple components and cannot be directly optimized, we substitute it with its upper bound $I(\boldsymbol{y}; \boldsymbol{z}^{(v)})$ based on its optimization direction. Therefore, under the dual-layer constraints at both the feature and category levels, the model for acquiring shared representations is obtained:

$$\min \frac{1}{|\mathcal{V}|} \sum_{v \in \mathcal{V}} \Big( - \underbrace{I(\boldsymbol{x}^{(v)}; \boldsymbol{z}) + I(\{\boldsymbol{x}^{\sim(v)}\}; \boldsymbol{z} \mid \boldsymbol{x}^{(v)})}_{\text{feature-level}}$$
$$+ \underbrace{I(\boldsymbol{x}^{(v)}; \boldsymbol{y}) - I(\boldsymbol{z}^{(v)}; \boldsymbol{y}) - I(\boldsymbol{y}; \boldsymbol{z}^{(v)}) + I(\boldsymbol{y}; \boldsymbol{z}^{(v)} \mid \{\boldsymbol{z}^{\sim(v)}\})}_{\text{category-level}} \Big). \qquad (4)$$

Due to the intractability of computing mutual information in high-dimensional spaces, we derive its bound that allows for reliable estimation to facilitate the optimization of model (4). For the first term $I(\boldsymbol{x}^{(v)}; \boldsymbol{z})$, its lower bound is typically expressed via a reconstruction loss, where $\boldsymbol{x}^{(v)}$ is decoded through the decoder $q^v(\boldsymbol{x}^{(v)}|\boldsymbol{z})$ to ensure the faithful preservation of the original view:

$$I(\boldsymbol{x}^{(v)}; \boldsymbol{z}) \geq \mathbb{E}_{p(\boldsymbol{x}^{(v)}; \boldsymbol{z})} \left[ \log q^v \left( \boldsymbol{x}^{(v)}|\boldsymbol{z} \right) \right] = \mathbb{E}_{\{\boldsymbol{x}\} \sim p(\{\boldsymbol{x}\})} \left[ \int p(\boldsymbol{z}|\{\boldsymbol{x}\}) \log q^v(\boldsymbol{x}^{(v)}|\boldsymbol{z}) d\boldsymbol{z} \right]. \quad (5)$$

Next, based on the definition of mutual information, the expansion for the second term is derived:

$$I(\{\boldsymbol{x}^{\sim(v)}\}; \boldsymbol{z} \mid \boldsymbol{x}^{(v)}) = \mathbb{E}_{p(\{\boldsymbol{x}^{\sim(v)}\}, \boldsymbol{z}, \boldsymbol{x}^{(v)})} \left[ \log \frac{p\left(\{\boldsymbol{x}^{\sim(v)}\}, \boldsymbol{z} \mid \boldsymbol{x}^{(v)}\right)}{p\left(\{\boldsymbol{x}^{\sim(v)}\} \mid \boldsymbol{x}^{(v)}\right) p\left(\boldsymbol{z} \mid \boldsymbol{x}^{(v)}\right)} \right]$$
$$= \int \int p(\{\boldsymbol{x}\}, \boldsymbol{z}) \log \frac{p(\boldsymbol{z}|\{\boldsymbol{x}\})}{p(\boldsymbol{z}|\boldsymbol{x}^{(v)})} d\{\boldsymbol{x}\} d\boldsymbol{z}. \qquad (6)$$

Since the distribution $p(\boldsymbol{z}|\{\boldsymbol{x}\})$ is difficult to obtain explicitly, we approximate it using a stochastic variational distribution $g^v(\boldsymbol{z}|\boldsymbol{x}^{(v)})$. Then, we can obtain the following transformation:

$$I(\{\boldsymbol{x}^{\sim(v)}\}; \boldsymbol{z} \mid \boldsymbol{x}^{(v)})$$
$$= \int \int p(\{\boldsymbol{x}\}, \boldsymbol{z}) \log \frac{p(\boldsymbol{z}|\{\boldsymbol{x}\})}{g^v(\boldsymbol{z}|\boldsymbol{x}^{(v)})} d\{\boldsymbol{x}\} d\boldsymbol{z} - \int p(\boldsymbol{x}^{(v)}) D_{KL} \left( p(\boldsymbol{z}|\boldsymbol{x}^{(v)}) || g^v(\boldsymbol{z}|\boldsymbol{x}^{(v)}) \right) d\boldsymbol{x}^{(v)}, \qquad (7)$$

where $D_{KL}(\cdot||\cdot)$ denotes the non-negative Kullback-Leibler divergence. Thus, the variational upper bound for $I(\{\boldsymbol{x}^{\sim(v)}\}; \boldsymbol{z} \mid \boldsymbol{x}^{(v)})$ can be established:

$$I(\{\boldsymbol{x}^{\sim(v)}\}; \boldsymbol{z} \mid \boldsymbol{x}^{(v)}) \leq \mathbb{E}_{\{\boldsymbol{x}\} \sim p(\{\boldsymbol{x}\})} \left[ D_{KL} \left( p(\boldsymbol{z}|\{\boldsymbol{x}\}) || g^v(\boldsymbol{z}|\boldsymbol{x}^{(v)}) \right) \right]. \qquad (8)$$

In summary , the trainable loss subject to the feature-level constraint is given by:

$$\mathcal{L}_f = \frac{1}{|\mathcal{V}|} \sum_{v \in \mathcal{V}} \left[ -\mathbb{E}_{\boldsymbol{z} \sim p(\boldsymbol{z}|\{\boldsymbol{x}\})} \log q^v(\boldsymbol{x}^{(v)}|\boldsymbol{z}) + D_{KL} \left( p(\boldsymbol{z}|\{\boldsymbol{x}\}) || g^v(\boldsymbol{z}|\boldsymbol{x}^{(v)}) \right) \right]. \qquad (9)$$

Under category-level constraints, the minimization of $I(\boldsymbol{x}^{(v)}; \boldsymbol{y}) - I(\boldsymbol{z}^{(v)}; \boldsymbol{y})$ is functionally equivalent to restricting $H(\boldsymbol{y}|\boldsymbol{z}^{(v)}) - H(\boldsymbol{y}|\boldsymbol{x}^{(v)})$, where $H(\cdot)$ denotes the Shannon entropy. Since the disparity in entropy is characterized by the divergence between distributions, the constraint objective naturally transitions to:

$$\min \sum_{v \in \mathcal{V}} D_{KL} \left( p(\boldsymbol{y}|\boldsymbol{z}^{(v)}) || p(\boldsymbol{y}|\boldsymbol{x}^{(v)}) \right) \tag{10}$$

Regarding the latter part of Model (4), the view-specific information is decomposed as follows:

$$I(\boldsymbol{y}; \boldsymbol{z}_i | \{\boldsymbol{z}^{\sim(v)}\}) = H(\boldsymbol{y} | \{\boldsymbol{z}^{\sim(v)}\}) - H(\boldsymbol{y} | \{\boldsymbol{z}\})$$
$$= -\int p(\boldsymbol{y} | \{\boldsymbol{z}^{\sim(v)}\}) \log p(\boldsymbol{y} | \{\boldsymbol{z}^{\sim(v)}\}) dy + \int p(\boldsymbol{y} | \{\boldsymbol{z}\}) \log p(\boldsymbol{y} | \{\boldsymbol{z}\}) dy, \tag{11}$$

Through term augmentation and subsequent expansion in logarithmic operations, we have

$$I(\boldsymbol{y}; \boldsymbol{z}^{(v)} | \{\boldsymbol{z}^{\sim(v)}\})$$
$$= -\int p(\boldsymbol{y} | \{\boldsymbol{z}^{\sim(v)}\}) \log \left[ \frac{p(\boldsymbol{y} | \{\boldsymbol{z}^{\sim(v)}\})}{p(\boldsymbol{y} | \{\boldsymbol{z}\})} \right] d\boldsymbol{y} - \int p(\boldsymbol{y} | \{\boldsymbol{z}^{\sim(v)}\}) \log p(\boldsymbol{y} | \{\boldsymbol{z}\}) \, d\boldsymbol{y}$$
$$+ \int p(\boldsymbol{y} | \{\boldsymbol{z}\}) \log \left[ \frac{p(\boldsymbol{y} | \{\boldsymbol{z}\})}{p(\boldsymbol{y} | \{\boldsymbol{z}^{\sim(v)}\})} \right] d\boldsymbol{y} + \int p(\boldsymbol{y} | \{\boldsymbol{z}\}) \log p(\boldsymbol{y} | \{\boldsymbol{z}^{\sim(v)}\}) d\boldsymbol{y}. \tag{12}$$
$$\leq D_{KL} \left( p(\boldsymbol{y} | \{\boldsymbol{z}\}) \parallel p(\boldsymbol{y} | \{\boldsymbol{z}^{\sim(v)}\}) \right) + H \left( p(\boldsymbol{y} | \{\boldsymbol{z}^{\sim(v)}\}), p(\boldsymbol{y} | \{\boldsymbol{z}\}) \right).$$

Owing to the congruent optimization objective of aligning $p(\boldsymbol{y}|\{\boldsymbol{z}\})$ and $p(\boldsymbol{y}|\{\boldsymbol{z}^{\sim(v)}\})$, we exclusively adopt $D_{KL} \left[ p(\boldsymbol{y}|\{\boldsymbol{z}\}) \parallel p(\boldsymbol{y}|\{\boldsymbol{z}^{\sim(v)}\}) \right]$ as the minimization target. Meanwhile, the existence of Eq. (3) enables the maximal elimination of view-specific noise to accentuate the label-related consensus information $I_v^s$. Then, the objective function guided by the category-level constraint is formulated as

$$\mathcal{L}_c = \frac{1}{|\mathcal{V}|} \sum_{v \in \mathcal{V}} \left( D_{KL} \left( p(\boldsymbol{y}|\boldsymbol{z}^{(v)}) || p(\boldsymbol{y}|\boldsymbol{x}^{(v)}) \right) + D_{KL} \left( p(\boldsymbol{y}|\{\boldsymbol{z}\}) \parallel p(\boldsymbol{y}|\{\boldsymbol{z}^{\sim(v)}\}) \right) \right). \tag{13}$$

## 2.3 ADAPTIVE VIEW FUSION AND LABEL REPRESENTATION LEARNING

The integration of view representations constitutes a critical challenge in multi-view learning. Given that variational inference optimizes the distribution of each view, we leverage the progressively refined distribution information to facilitate view fusion. Within the network architecture, each view is processed through two encoders to estimate the latent distribution of its shared representations. Specifically, the distribution is modeled as $p(\boldsymbol{z}^{(v)} \mid \boldsymbol{x}^{(v)}) := \mathcal{N}(f_\mu^v(\boldsymbol{x}^{(v)}), f_{\sigma^2}^v(\boldsymbol{x}^{(v)})\boldsymbol{I})$, where $f_\mu^v$ and $f_{\sigma^2}^v$ are the mean and variance encoders. To ensure that the shared feature incorporates information from all views and benefits from the greater stability of representations with lower variance, we adopt the product-of-experts (PoE) framework (Hinton, 2002) with the one-vote property to perform weighted fusion of the distribution parameters across views:

$$\begin{cases} \mu_s = \dfrac{\sum_{v \in \mathcal{V}} f_\mu^v(\boldsymbol{x}^{(v)}) \frac{1}{f_{\sigma^2}^v(\boldsymbol{x}^{(v)})}}{\sum_{v \in \mathcal{V}} \frac{1}{f_{\sigma^2}^v(\boldsymbol{x}^{(v)})} + 1} \\ \sigma_s^2 = \dfrac{1}{\sum_{v \in \mathcal{V}} \frac{1}{f_{\sigma^2}^v(\boldsymbol{x}^{(v)})} + 1}. \end{cases} \tag{14}$$

Then, we employ the reparameterization trick to sample $S$ times from the distribution:

$$\boldsymbol{z} = \frac{1}{S} \sum_{i=1}^{S} \left( \mu_s + \sigma_s \odot \delta^i \right), \tag{15}$$

where $\delta^i \in \mathbb{R}^d$ denotes the $i$-th sampling from the standard Gaussian distribution and $\odot$ indicates element-wise multiplication. The representations $\{\boldsymbol{z}^{(v)}\}_{v=1}^V$ extracted from each view are also sampled from their respective distributions following Eq. (15). During view fusion, it is essential to not only account for the aggregation of representation information but also to incorporate the impact

of label information. Since multiple labels are typically encoded as one-hot vectors, which lacks the flexibility to capture label semantics, particularly in scenarios with missing labels. To address this, we adopt a data-driven approach to introduce label prototypes, ensuring that the semantic information carried by these prototypes is closely aligned with the ground truth labels. In order to explicitly model label expressions, we employ stochastic encoders to fit the underlying distribution $\mathcal{N}(\mu_i, \sigma_i^2 \boldsymbol{I})$ for each label prototype, where $\mu_i$ and $\sigma_i^2$ are the $d$-dimensional mean and variance outputs, respectively, produced by the encoders $h_\mu(\boldsymbol{b}_i)$ and $h_{\sigma^2}(\boldsymbol{b}_i)$. $\boldsymbol{b}_i \in \mathbb{R}^C$ serves as a learnable embedding corresponding to the $i$-th class, which is initialized as a one-hot vector with the $i$-th entry is 1. After obtaining $\{\boldsymbol{l}_i\}_{i=1}^C$ through stochastic sampling, it is necessary to capture the intricate correlations between these label representations, which forms a crucial determinant in enhancing the performance of multi-label classification. Considering that the manifestation of label correlations differs across samples, we adopt a nuanced approach that centers on instance-level relevance to strengthen the similarity between the cross-view representation of each sample and the label attributes it possesses. Specifically, we sample the shared feature $\boldsymbol{z}$ according to Eq. (15), with its associated known label prototype being $\{\boldsymbol{l}_i | i \in \mathcal{U}\}$. By using the cosine similarity as the criterion, the alignment loss designed to capture label correlations is as follows:

$$\mathcal{L}_a = \frac{1}{|\mathcal{U}|} \sum_{i \in \mathcal{U}} \frac{\langle \boldsymbol{z} \cdot \boldsymbol{l}_i \rangle}{\|\boldsymbol{z}\| \|\boldsymbol{l}_i\|}. \tag{16}$$

By optimizing loss $\mathcal{L}_a$, we refine the mapping semantic between features and labels, while simultaneously highlighting the associations among label prototypes, which are tailored for application to each individual sample. Subsequently, during label prediction, it is important to synthesize the generalized information from multiple views with the semantic representations of individual categories. When these information exhibit coherence, it becomes feasible to infer that the sample contains the relevant labels. To this end, we utilize a neural network to adaptively gauge the degree of similarity between view representations and category embeddings:

$$\boldsymbol{p}_i^0 = \omega \left( g_c \left( \boldsymbol{z} \oplus \boldsymbol{l}_i \right) \right), \tag{17}$$

where $g_c$ is a fully connected layer, $\oplus$ denotes concatenation operation and $\sigma_S$ is the Sigmoid activation function. The derivation of $\boldsymbol{p}_i^0$ solely relies on the shared representation $\boldsymbol{z}$ resulting from the fusion of feature information. To further refine the integration of effective multimodal information, we incorporate multi-label semantic information into the fusion process. To achieve this, we propose a label-guided post-view fusion framework, where $\boldsymbol{p}^0$ and the label distributions $p(\boldsymbol{y}|\{\boldsymbol{z}^{\sim(v)}\})$ obtained from the exclusion of each view are adaptively merged. This strategy is designed to mitigate the adverse effects of heterogeneous views on label recognition while preserving the most discriminative feature information, thereby improving the reliability of the prediction outcome. Then, we utilize the computed result $\mathcal{L}_{con} = \frac{1}{|\mathcal{U}|} \sum_{i \in \mathcal{U}} \left( \boldsymbol{p}_i^2 + (1 - \boldsymbol{p}_i)^2 \right)$ of the predicted label distribution as its confidence measure. Besides, we derive $\mathcal{L}_{con}^0$ based on $\boldsymbol{p}^0$, and calculate $\mathcal{L}_{con}^{(v)}$ from $p(\boldsymbol{y}|\{\boldsymbol{z}^{\sim(v)}\})$. The formulation of $\mathcal{L}_{con}$ indicates that the value of 0.5 serves as the classification boundary. Scores significantly exceeding 0.5 indicate a stronger tendency toward positive labels, while those substantially below 0.5 reflect an increased likelihood of negative assignment. Therefore, by employing $\mathcal{L}_{con}$ as the weighting factor for late fusion, we can obtain the enhanced result as the final prediction:

$$\boldsymbol{p}_i^t = \sum_{v \in \mathcal{V}} \mathcal{L}_{con}^{(v)} p(\boldsymbol{y}_i | \{\boldsymbol{z}^{\sim(v)}\}) + \mathcal{L}_{con}^{(0)} \boldsymbol{p}_i^0, \tag{18}$$

where all weighting coefficients $\mathcal{L}_{con}^{(v)} (0 \le v \le V)$ are normalized in advance. To enhance the classification discriminability and reinforce the interaction term $I(\boldsymbol{y}; \boldsymbol{z}^{(v)})$ in model (4), we employ the following cross-entropy loss:

$$\mathcal{L}_{BCE} = \frac{1}{|\mathcal{U}|} \sum_{i \in \mathcal{U}} \left[ \boldsymbol{y}_i \log \boldsymbol{p}_i + (1 - \boldsymbol{y}_i) \log (1 - \boldsymbol{p}_i) \right]. \tag{19}$$

The classification loss in our method is the aggregation of four distinct components, with one arising from the final prediction and the remaining three emanating from pseudo-predictions $p(\boldsymbol{y}_i | \{\boldsymbol{z}^{\sim(v)}\})$, $p(\boldsymbol{y}_i | \boldsymbol{x}^{(v)})$, and $p(\boldsymbol{y}_i | \boldsymbol{z}^{(v)})$, which collectively constitutes the overall loss $\mathcal{L}_{BCE}^t$. Thus, the total training loss of TITRR is as below:

$$\mathcal{L} = \mathcal{L}_{BCE}^t + \mathcal{L}_a + \lambda_1 \mathcal{L}_f + \lambda_2 \mathcal{L}_c, \tag{20}$$

where $\lambda_1$ and $\lambda_2$ govern the trade-off between the empirical values and impacts of different losses.

## 3 EXPERIMENTS

### 3.1 DATASETS AND METRICS

In our experiments, we employ six widely used multi-view multi-label datasets to evaluate the effectiveness of our method, i.e., Corel 5k Duygulu et al. (2002), ESPGame Ahn & Dabbish (2004), IAPRTC12 Grubinger et al. (2006), Mirflickr Huiskes & Lew (2008), Pascal07 Everingham et al. (2010), and OBJECT Hao et al. (2024). Following the evaluation protocols in Liu et al. (2023b); Wen et al. (2023), we adopt the following six metrics to form a comprehensive assessment framework, i.e., Hamming Loss (HL), Ranking Loss (RL), OneError (OE), Coverage (Cov), Average Precision (AP), and Area Under Curve (AUC). For clarity in comparison, we report 1-HL, 1-OE, 1-Cov, and 1-RL, where higher values consistently indicate better performance.

### 3.2 COMPARISON METHODS

To assess the performance of our method, we compare it with nine state-of-the-art approaches, i.e., AIMNet Liu et al. (2024a), DICNet Liu et al. (2023b), DIMC Wen et al. (2023), iMVWL Tan et al. (2018), LMVCAT Liu et al. (2023c), MTD Liu et al. (2023a), SIP Liu et al. (2024b), LVSL Zhao et al. (2022), and DM2L Ma & Chen (2021). The first seven methods are capable of simultaneously handling missing views and labels. Since LVSL cannot directly process incomplete data, we impute missing views using the mean of available instances and fill absent labels with zeros. DM2L is a kernel-based nonlinear method for incomplete multi-label learning. Thus, we concatenate all recovered views into a single representation to apply DM2L. All hyperparameters of the compared methods are set according to the recommended configurations in their original implementations, ensuring a fair and reproducible comparison.

### 3.3 IMPLEMENTATION DETAILS

To simulate partial view scenarios, a proportion of instances determined by the Partial Example Ratio (PER) are randomly masked in each view, while ensuring each sample retains at least one complete view. For weak supervision, label omissions are applied to both positive and negative tags according to the Label Missing Ratio (LMR). Incomplete data construction is repeated multiple times to mitigate randomness. Datasets are split into training, validation, and test sets with a 7:1:2 ratio. Our method is implemented in PyTorch and trained on an NVIDIA GeForce RTX 4090 GPU.

### 3.4 EXPERIMENTAL RESULTS AND ANALYSIS

To rigorously evaluate the effectiveness of TITRL in handling absent views and incomplete labels, we conduct extensive comparative experiments against nine representative algorithms across six benchmark datasets under varying levels of data sparsity. Specifically, the proportions of missing views (PER) and labels (LMR) are set to $\{30\%, 50\%, 70\%, 90\%\}$. The results in terms of the mean and standard deviation at PER=50% and LMR=50% are summarized in Table 1, along with the average ranking across six evaluation metrics to provide an aggregated performance assessment. In addition, Fig. 2 visualizes how AP evolves as the missing proportion increases, while Fig. 3 presents radar plots that jointly capture multi-metric performance distribution at PER=90% and LMR=90%. These results collectively provide a holistic evaluation of predictive accuracy and model robustness.

From the comparison results, several important observations can be drawn: (i) TITRL consistently secures the best results across almost all datasets and metrics. For instance, on Corel5k, TITRL achieves an AP score of 0.432, outperforming SIP (0.414) and MTD (0.410), with similar margins observed on other datasets. Besides, TITRL maintains its superiority on large-scale datasets under data missingness like ESPGame and Mirflickr, which underscores its scalability and resilience. (ii) Drawn from Fig. 2, we can find that while competing methods suffer steep performance degradation when missing ratios achieve a high level, TITRL continues to exhibit considerable performance. For example, when PER=70%, the performance of all comparison methods stays below 0.36, while TITRL surpasses 0.4 by a certain margin. Although our method still outperforms others under conditions of severe label missingness, the performance gap is prominently reflected in the presence of feature absent. This further highlights the critical importance of multi-view representation learning, a role that our method is well equipped to fulfill. As depicted in the radar chart of Fig. 3, it is evident that our method consistently occupies the outermost boundary, which indicates that TITRL stands out even under highly challenging conditions across various evaluation perspectives. Consequently, our method demonstrates strong robustness in addressing the problem of data incompleteness and

shows significant potential for broader adoption. (iii) As evidenced by Table 1, our method consistently maintains the top position, while the rankings of alternative approaches remain volatile, which shows the exceptional performance stability of our method. Against top competing methods SIP and MTD, our approach also demonstrates superiority, which underscores the pivotal role of introducing label integration strategies in the feature extraction process and fine-grained characterization of label semantics. Compared with the deep learning–based methods AIMNet, DICNet and DIMC, the substantial advantage of TITRL further reveals the importance of jointly considering the view property and label information during the fusion process.

Table 1: Experimental results of nine methods on the six datasets with $50\%$ PER and $50\%$ LMR. 'AVE' refers to the mean ranking of the corresponding method across all six metrics. The best and second best results are highlighted in red and blue, respectively.

| DATA | METRIC | AIMNet | DICNet | DIMC | DM2L | iMVWL | LMVCAT | LVSL | MTD | SIP | TITRL |
|---|---|---|---|---|---|---|---|---|---|---|---|
| COR | 1-HL | $0.988_{0.000}$ | $0.987_{0.000}$ | $0.987_{0.000}$ | $0.987_{0.000}$ | $0.978_{0.000}$ | $0.986_{0.000}$ | $0.987_{0.000}$ | $0.988_{0.000}$ | $0.988_{0.000}$ | $0.988_{0.000}$ |
| | 1-OE | $0.478_{0.011}$ | $0.460_{0.012}$ | $0.446_{0.009}$ | $0.378_{0.014}$ | $0.308_{0.017}$ | $0.448_{0.011}$ | $0.353_{0.017}$ | $0.492_{0.011}$ | $0.492_{0.014}$ | $0.509_{0.014}$ |
| | 1-Cov | $0.766_{0.004}$ | $0.726_{0.007}$ | $0.709_{0.008}$ | $0.640_{0.007}$ | $0.701_{0.003}$ | $0.720_{0.004}$ | $0.720_{0.005}$ | $0.754_{0.004}$ | $0.781_{0.004}$ | $0.795_{0.006}$ |
| | 1-RL | $0.900_{0.002}$ | $0.881_{0.004}$ | $0.874_{0.004}$ | $0.843_{0.004}$ | $0.864_{0.002}$ | $0.876_{0.004}$ | $0.879_{0.002}$ | $0.893_{0.004}$ | $0.908_{0.003}$ | $0.914_{0.003}$ |
| | AP | $0.404_{0.005}$ | $0.381_{0.006}$ | $0.370_{0.005}$ | $0.318_{0.005}$ | $0.281_{0.005}$ | $0.379_{0.006}$ | $0.311_{0.005}$ | $0.410_{0.007}$ | $0.414_{0.006}$ | $0.432_{0.007}$ |
| | AUC | $0.903_{0.002}$ | $0.883_{0.004}$ | $0.877_{0.004}$ | $0.846_{0.004}$ | $0.867_{0.002}$ | $0.879_{0.004}$ | $0.882_{0.002}$ | $0.896_{0.004}$ | $0.910_{0.002}$ | $0.916_{0.002}$ |
| | AVE | 3.5 | 5.0 | 7.2 | 9.0 | 9.5 | 6.8 | 7.3 | 3.2 | 2.2 | 1.0 |
| ESP | 1-HL | $0.983_{0.000}$ | $0.983_{0.000}$ | $0.983_{0.000}$ | $0.983_{0.000}$ | $0.972_{0.000}$ | $0.982_{0.000}$ | $0.983_{0.000}$ | $0.983_{0.000}$ | $0.983_{0.000}$ | $0.983_{0.000}$ |
| | 1-OE | $0.442_{0.006}$ | $0.440_{0.006}$ | $0.431_{0.009}$ | $0.302_{0.008}$ | $0.343_{0.010}$ | $0.432_{0.006}$ | $0.365_{0.006}$ | $0.452_{0.007}$ | $0.450_{0.006}$ | $0.481_{0.006}$ |
| | 1-Cov | $0.621_{0.003}$ | $0.601_{0.003}$ | $0.586_{0.004}$ | $0.532_{0.003}$ | $0.548_{0.004}$ | $0.587_{0.003}$ | $0.578_{0.002}$ | $0.617_{0.004}$ | $0.622_{0.003}$ | $0.631_{0.008}$ |
| | 1-RL | $0.845_{0.002}$ | $0.836_{0.002}$ | $0.830_{0.002}$ | $0.804_{0.002}$ | $0.807_{0.002}$ | $0.827_{0.002}$ | $0.829_{0.001}$ | $0.843_{0.002}$ | $0.847_{0.002}$ | $0.852_{0.004}$ |
| | AP | $0.306_{0.003}$ | $0.300_{0.003}$ | $0.294_{0.003}$ | $0.229_{0.003}$ | $0.243_{0.004}$ | $0.293_{0.003}$ | $0.266_{0.003}$ | $0.309_{0.003}$ | $0.309_{0.003}$ | $0.339_{0.003}$ |
| | AUC | $0.850_{0.001}$ | $0.841_{0.002}$ | $0.835_{0.002}$ | $0.808_{0.001}$ | $0.813_{0.002}$ | $0.832_{0.001}$ | $0.834_{0.001}$ | $0.847_{0.002}$ | $0.851_{0.002}$ | $0.855_{0.003}$ |
| | AVE | 3.7 | 4.5 | 5.7 | 9.7 | 9.2 | 7.3 | 7.2 | 3.5 | 2.3 | 1.0 |
| IAP | 1-HL | $0.981_{0.000}$ | $0.981_{0.000}$ | $0.981_{0.000}$ | $0.980_{0.000}$ | $0.969_{0.000}$ | $0.980_{0.000}$ | $0.981_{0.000}$ | $0.981_{0.000}$ | $0.981_{0.000}$ | $0.982_{0.000}$ |
| | 1-OE | $0.457_{0.008}$ | $0.464_{0.006}$ | $0.454_{0.006}$ | $0.378_{0.008}$ | $0.351_{0.008}$ | $0.433_{0.009}$ | $0.377_{0.007}$ | $0.479_{0.007}$ | $0.459_{0.006}$ | $0.508_{0.008}$ |
| | 1-Cov | $0.675_{0.004}$ | $0.649_{0.005}$ | $0.630_{0.005}$ | $0.556_{0.005}$ | $0.565_{0.004}$ | $0.646_{0.004}$ | $0.605_{0.004}$ | $0.670_{0.004}$ | $0.678_{0.003}$ | $0.693_{0.006}$ |
| | 1-RL | $0.884_{0.001}$ | $0.874_{0.002}$ | $0.868_{0.002}$ | $0.837_{0.002}$ | $0.833_{0.002}$ | $0.868_{0.002}$ | $0.857_{0.002}$ | $0.882_{0.002}$ | $0.886_{0.001}$ | $0.893_{0.002}$ |
| | AP | $0.329_{0.003}$ | $0.326_{0.003}$ | $0.318_{0.002}$ | $0.254_{0.002}$ | $0.236_{0.002}$ | $0.313_{0.004}$ | $0.262_{0.002}$ | $0.340_{0.002}$ | $0.331_{0.002}$ | $0.377_{0.004}$ |
| | AUC | $0.885_{0.001}$ | $0.876_{0.002}$ | $0.870_{0.001}$ | $0.838_{0.001}$ | $0.835_{0.001}$ | $0.870_{0.002}$ | $0.859_{0.001}$ | $0.883_{0.002}$ | $0.887_{0.001}$ | $0.894_{0.002}$ |
| | AVE | 4.0 | 4.3 | 6.0 | 8.8 | 8.8 | 6.8 | 8.0 | 3.0 | 2.8 | 1.0 |
| MIR | 1-HL | $0.890_{0.001}$ | $0.890_{0.001}$ | $0.890_{0.001}$ | $0.876_{0.001}$ | $0.840_{0.004}$ | $0.880_{0.004}$ | $0.877_{0.001}$ | $0.893_{0.001}$ | $0.890_{0.001}$ | $0.896_{0.001}$ |
| | 1-OE | $0.646_{0.009}$ | $0.647_{0.010}$ | $0.646_{0.008}$ | $0.533_{0.008}$ | $0.511_{0.016}$ | $0.639_{0.009}$ | $0.609_{0.007}$ | $0.667_{0.006}$ | $0.654_{0.007}$ | $0.683_{0.006}$ |
| | 1-Cov | $0.673_{0.003}$ | $0.662_{0.004}$ | $0.657_{0.003}$ | $0.615_{0.002}$ | $0.588_{0.013}$ | $0.665_{0.002}$ | $0.624_{0.002}$ | $0.681_{0.002}$ | $0.669_{0.006}$ | $0.688_{0.003}$ |
| | 1-RL | $0.874_{0.002}$ | $0.869_{0.003}$ | $0.867_{0.003}$ | $0.835_{0.001}$ | $0.809_{0.011}$ | $0.862_{0.003}$ | $0.847_{0.001}$ | $0.878_{0.001}$ | $0.873_{0.002}$ | $0.886_{0.002}$ |
| | AP | $0.599_{0.003}$ | $0.595_{0.007}$ | $0.592_{0.006}$ | $0.519_{0.001}$ | $0.495_{0.017}$ | $0.589_{0.004}$ | $0.548_{0.003}$ | $0.614_{0.004}$ | $0.603_{0.005}$ | $0.629_{0.004}$ |
| | AUC | $0.861_{0.001}$ | $0.855_{0.002}$ | $0.854_{0.002}$ | $0.828_{0.001}$ | $0.801_{0.017}$ | $0.852_{0.003}$ | $0.839_{0.001}$ | $0.864_{0.001}$ | $0.859_{0.002}$ | $0.871_{0.002}$ |
| | AVE | 3.8 | 4.7 | 6.2 | 9.0 | 10.0 | 6.7 | 8.0 | 2.0 | 3.5 | 1.0 |
| OBJ | 1-HL | $0.948_{0.001}$ | $0.948_{0.001}$ | $0.947_{0.001}$ | $0.935_{0.001}$ | $0.899_{0.002}$ | $0.940_{0.002}$ | $0.935_{0.001}$ | $0.949_{0.001}$ | $0.948_{0.001}$ | $0.950_{0.001}$ |
| | 1-OE | $0.619_{0.015}$ | $0.601_{0.011}$ | $0.594_{0.012}$ | $0.537_{0.011}$ | $0.465_{0.018}$ | $0.604_{0.016}$ | $0.450_{0.008}$ | $0.627_{0.011}$ | $0.626_{0.009}$ | $0.648_{0.008}$ |
| | 1-Cov | $0.807_{0.006}$ | $0.794_{0.006}$ | $0.793_{0.006}$ | $0.768_{0.005}$ | $0.744_{0.006}$ | $0.796_{0.006}$ | $0.759_{0.006}$ | $0.813_{0.006}$ | $0.809_{0.006}$ | $0.818_{0.006}$ |
| | 1-RL | $0.888_{0.005}$ | $0.876_{0.004}$ | $0.875_{0.004}$ | $0.860_{0.004}$ | $0.833_{0.006}$ | $0.878_{0.006}$ | $0.850_{0.004}$ | $0.890_{0.004}$ | $0.889_{0.004}$ | $0.897_{0.003}$ |
| | AP | $0.639_{0.010}$ | $0.627_{0.009}$ | $0.623_{0.010}$ | $0.577_{0.008}$ | $0.512_{0.014}$ | $0.630_{0.012}$ | $0.537_{0.008}$ | $0.649_{0.009}$ | $0.649_{0.008}$ | $0.665_{0.006}$ |
| | AUC | $0.897_{0.004}$ | $0.886_{0.004}$ | $0.885_{0.004}$ | $0.872_{0.004}$ | $0.846_{0.006}$ | $0.888_{0.006}$ | $0.864_{0.004}$ | $0.900_{0.004}$ | $0.898_{0.005}$ | $0.906_{0.003}$ |
| | AVE | 4.0 | 5.7 | 6.8 | 8.2 | 9.8 | 5.3 | 9.0 | 2.0 | 3.0 | 1.0 |
| PAS | 1-HL | $0.931_{0.001}$ | $0.931_{0.000}$ | $0.931_{0.001}$ | $0.927_{0.001}$ | $0.882_{0.004}$ | $0.915_{0.005}$ | $0.928_{0.001}$ | $0.933_{0.001}$ | $0.932_{0.001}$ | $0.935_{0.001}$ |
| | 1-OE | $0.462_{0.010}$ | $0.443_{0.007}$ | $0.435_{0.010}$ | $0.419_{0.006}$ | $0.366_{0.039}$ | $0.433_{0.016}$ | $0.418_{0.008}$ | $0.474_{0.008}$ | $0.468_{0.008}$ | $0.496_{0.009}$ |
| | 1-Cov | $0.781_{0.007}$ | $0.749_{0.005}$ | $0.738_{0.010}$ | $0.720_{0.004}$ | $0.674_{0.011}$ | $0.759_{0.006}$ | $0.738_{0.003}$ | $0.790_{0.006}$ | $0.778_{0.004}$ | $0.795_{0.005}$ |
| | 1-RL | $0.830_{0.006}$ | $0.804_{0.002}$ | $0.792_{0.008}$ | $0.778_{0.003}$ | $0.736_{0.011}$ | $0.808_{0.006}$ | $0.797_{0.002}$ | $0.836_{0.004}$ | $0.828_{0.004}$ | $0.844_{0.004}$ |
| | AP | $0.549_{0.007}$ | $0.517_{0.004}$ | $0.510_{0.008}$ | $0.482_{0.005}$ | $0.438_{0.022}$ | $0.524_{0.009}$ | $0.486_{0.005}$ | $0.562_{0.005}$ | $0.552_{0.006}$ | $0.581_{0.007}$ |
| | AUC | $0.851_{0.005}$ | $0.827_{0.004}$ | $0.817_{0.008}$ | $0.806_{0.003}$ | $0.767_{0.011}$ | $0.830_{0.006}$ | $0.823_{0.002}$ | $0.855_{0.006}$ | $0.848_{0.005}$ | $0.861_{0.003}$ |
| | AVE | 3.5 | 5.7 | 7.2 | 8.7 | 10.0 | 6.0 | 7.5 | 2.0 | 3.5 | 1.0 |

## 3.5 ABLATION STUDY

Table 2: Ablation study on Pascal07, OBJECT and Mirflickr with PER=50% and LMR=50%. '✓' and '✗' represent the used and not used corresponding item, respectively.

| $S_1$ | $S_2$ | $S_3$ | Pascal07 | | | | OBJECT | | | | Mirflickr | | | |
|---|---|---|---|---|---|---|---|---|---|---|---|---|---|---|
| | | | AP | AUC | 1-RL | 1-OE | AP | AUC | 1-RL | 1-OE | AP | AUC | 1-RL | 1-OE |
| ✗ | ✓ | ✓ | 0.580 | 0.858 | 0.843 | 0.492 | 0.664 | 0.902 | 0.893 | 0.642 | 0.623 | 0.862 | 0.878 | 0.681 |
| ✓ | ✗ | ✓ | 0.561 | 0.845 | 0.834 | 0.476 | 0.652 | 0.896 | 0.889 | 0.632 | 0.612 | 0.851 | 0.865 | 0.665 |
| ✓ | ✓ | ✗ | 0.584 | 0.861 | 0.847 | 0.495 | 0.668 | 0.905 | 0.897 | 0.646 | 0.628 | 0.866 | 0.881 | 0.685 |
| ✓ | ✓ | ✓ | **0.588** | **0.867** | **0.851** | **0.501** | **0.673** | **0.910** | **0.901** | **0.651** | **0.633** | **0.870** | **0.885** | **0.690** |

The ablation studies are carried out to deeply examine the the effect of the three key modules in TITRL, i.e., a dual-layer constraint framework for shared representation learning ($S_1$), the strategy of view fusion guided by label information ($S_2$), and sample-level label correlation semantic learning. After the separate removal of $S_1$, $S_2$, and $S_3$, losses $\mathcal{L}_f$ and $\mathcal{L}_c$ tied to representation extraction are omitted, view aggregation is realized solely through distributed fusion, and loss $\mathcal{L}_a$ is excluded without accounting for label dependencies, respectively. The ablation results shown in Table 2 lead

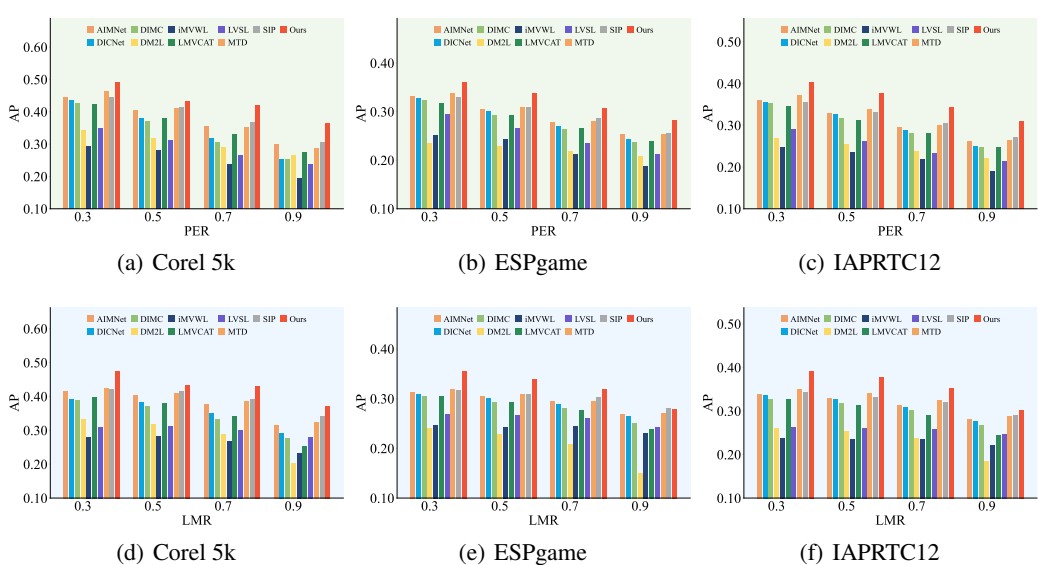

Figure 2: Experimental results on three datasets with one of PER and LMR fixed at 50% and the other varying from 30% to 90%

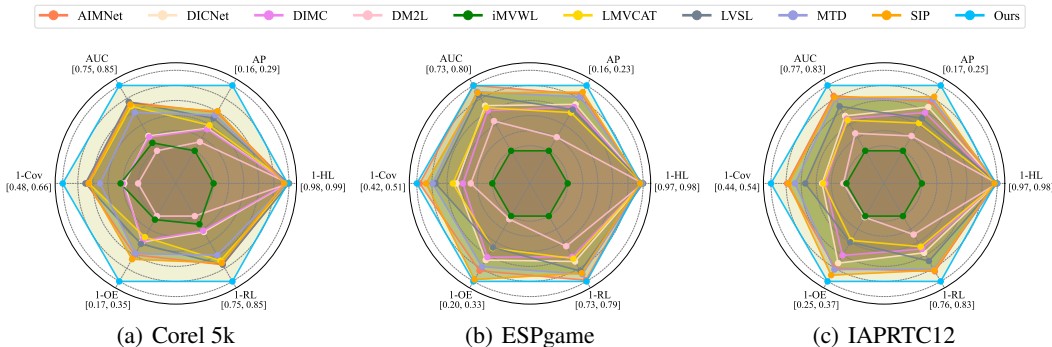

Figure 3: Experimental results of ten methods on three datasets with PER= 90% and LMR= 90%.

to the following findings: (i) The performance degradation observed upon the removal of any single module highlights the deliberate design of TITRL. (ii) Incorporating label semantics into the fusion process is instrumental in enhancing classification performance, as it enables the selective integration of the discriminative information from view representations. The training loss that facilitates the learning of both the shared representation and label correlation semantics exerts a positive influence, showing that our approach is valuable for advancing semantic exploration at both levels.

## 4 CONCLUSION

In this paper, we propose a Theory-Inspired Task-Relevant Representation Learning method called TITRL to address the IMvMLC problem, which is driven by the imperative to purify shared representations, improve the reliability of view fusion, and accurately capture multi-label correlation semantics. Specifically, TITRL introduces a dual-layer constraint framework based on the interaction of mutual information, which enables the disentanglement of view-specific noise. By deriving the tractable variational bounds, we provide theoretical guidance for learning pure shared information in a principled manner. For view fusion, TITRL integrates a distribution-aware strategy that leverages the statistical view property with a confidence-driven late fusion mechanism, thereby reinforcing the stability of representation expression and predictive reliability. Furthermore, we explicitly model sample-level label correlations by aligning shared representations with learnable label prototypes, allowing for flexible use of label dependencies. Extensive experiments on multiple benchmark datasets demonstrate TITRL's superiority and robustness, particularly in high-deficiency conditions where most baselines fail.

ETHICS STATEMENT

We confirm that our work adheres to the ICLR Code of Ethics. This research does not involve human subjects or raise concerns regarding privacy, legal compliance, or harmful insights. The datasets used are publicly available and properly cited. No conflicts of interest or external sponsorships influenced the research. We are committed to maintaining research integrity throughout the process.

REPRODUCIBILITY STATEMENT

We have taken several steps to ensure the reproducibility of our work. The detailed description of the methodology, experimental setup, and datasets used can be found in the main text and supplementary materials. The key components required for reproducing the experiments include the model architecture, training procedure, and evaluation metrics, all of which are explicitly described. In addition, for the reproducibility of the novel models and algorithms, we will release the source code upon acceptance. For the theoretical claims, we provide a comprehensive explanation of the underlying assumptions and the complete proofs of the claims in the appendix. Furthermore, all datasets used in the experiments are publicly available, and a detailed description of the data processing steps, including handling missing views and labels, can be found in the supplementary materials. We encourage the readers to refer to the provided supplementary materials for any further details required to replicate the results.

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

## A APPENDIX

### A.1 RELATED WORKS

#### A.1.1 MVMLC

The integration of multi-view learning with multi-label classification introduces algorithmic complexity but enables a comprehensive representation of object attributes. Consequently, algorithms in this domain were designed to address multi-view processing and multi-label recognition concurrently. For instance, LSA-MML Zhang et al. (2018) developed a shared latent representation via matrix factorization and aligns latent basis matrices across views in the kernel space using the Hilbert-Schmidt Independence Criterion (HSIC) to maximize inter-view dependency. Wu et al. proposed SIMM Wu et al. (2019), a deep MvMLC network that learned a shared subspace and view-specific features, ensuring a clear separation between shared and specific representations to capture the distinct contribution of each view. CDMM Zhao et al. (2021) focused on label relevance by promoting inter-view consistency and diversity through a straightforward approach, constructing a label affinity matrix that integrates Jaccard similarity in the label space with instance-level proximity in the feature space, followed by label propagation to enhance label representation. LVSL Zhao et al. (2022) addressed non-aligned multi-view multi-label classification by jointly exploring view-specific labels and low-rank label structures while preserving geometric properties of the original data via Laplacian graph regularization.

#### A.1.2 IMVMLC

Incomplete multi-view multi-label learning has attracted growing attention due to challenges posed by missing data in both feature and label domains. Missing views are generally addressed either by disregarding them with prior knowledge or reconstructing them through data completion. Liu et al. Liu et al. (2022) introduced a missing indicator matrix that enabled adaptively handling instances with missing views, while Wen et al. Wen et al. (2019) proposed UEAF, which reconstructed missing views using an error matrix informed by local structure and enhanced alignment through reverse graph learning. The challenge of incomplete labels has inspired multiple solutions. DM2L Ma & Chen (2021) inferred latent labels from partial annotations, integrated label correlations and low-rank constraints for robust feature learning, and refined pseudo-labels via self-supervised learning. GLOCAL Zhu et al. (2017) constructed bidirectional graphs between labels and instances to capture global label semantics under missing annotations, incorporated local priors and structure-preserving constraints to mitigate label noise and enhance feature discriminability. The methods capable of simultaneously handling missing views and labels initially involved the exploration of some traditional approaches. For example, iMVWL Tan et al. (2018) handled missing views and labels by jointly learning a shared subspace, a weak-label predictor, and a low-rank label correlation matrix. NAIM3L captured global high-rank and local low-rank structures of multi-label matrix and employed an efficient ADMM algorithm with linear time complexity for large-scale data. However, the performance of these methods is limited by their capacity to extract shallow representations and perform linear predictions.

Recent deep learning-based methods have achieved substantial improvements. DIMC Wen et al. (2023) utilizes an end-to-end network to extract high-level discriminative features with decoder-based reconstruction for robustness. DICNet Liu et al. (2023b) applied instance-level contrastive learning to unify representations of identical samples across views. LMVCAT Liu et al. (2023c) employed a Transformer-based framework with multi-head attention to facilitate feature interactions and category-aware modules to capture label semantics. Additional methods, including AIMNet Liu et al. (2024a), SIP Liu et al. (2024b), and MTD Liu et al. (2023a), explored attention-guided embedding completion, information bottleneck principles, and masked dual-channel decoupling strategies to reconstruct missing views, derive shared information and capture view-specific information, respectively. Together, these approaches emphasize the significance of extracting advanced representations, preserving label correlations, and utilizing sophisticated network architectures to address incomplete multi-view multi-label learning challenges.

## A.2 A COMPLETE DERIVATION OF THE TASK-RELEVANT REPRESENTATION LEARNING MODEL UNDER A DUAL-LAYER CONSTRAINT

In this section, we present a detailed derivation of model (4):

$$
\min \frac{1}{|\mathcal{V}|} \sum_{v \in \mathcal{V}} \Big( - \underbrace{I(\boldsymbol{x}^{(v)}; \boldsymbol{z}) + I(\{\boldsymbol{x}^{\sim(v)}\}; \boldsymbol{z} \mid \boldsymbol{x}^{(v)})}_{\text{feature-level}}
$$
$$
+ \underbrace{I(\boldsymbol{x}^{(v)}; \boldsymbol{y}) - I(\boldsymbol{z}^{(v)}; \boldsymbol{y}) - I(\boldsymbol{y}; \boldsymbol{z}^{(v)}) + I(\boldsymbol{y}; \boldsymbol{z}^{(v)} \mid \{\boldsymbol{z}^{\sim(v)}\})}_{\text{category-level}} \Big).
$$

(21)

For the first term, we proceed with the following expansion based on the definition of mutual information:

$$
I(\boldsymbol{x}^{(v)}; \boldsymbol{z}) = \int \int p(\boldsymbol{x}^{(v)}, \boldsymbol{z}) \log \frac{p(\boldsymbol{x}^{(v)}, \boldsymbol{z})}{p(\boldsymbol{x}^{(v)}) p(\boldsymbol{z})} d\boldsymbol{z} d\boldsymbol{x}^{(v)}.
$$

(22)

Considering $p(\boldsymbol{x}^{(v)}, \boldsymbol{z}) = p(\boldsymbol{x}^{(v)}|\boldsymbol{z}) p(\boldsymbol{x}^{(v)})$, we have

$$
\begin{aligned}
I(\boldsymbol{x}^{(v)}; \boldsymbol{z}) &= \int \int p(\boldsymbol{x}^{(v)}|\boldsymbol{z}) p(\boldsymbol{x}^{(v)}) \log \frac{p(\boldsymbol{x}^{(v)}|\boldsymbol{z}) p(\boldsymbol{x}^{(v)})}{p(\boldsymbol{x}^{(v)}) p(\boldsymbol{z}|\boldsymbol{x}^{(v)})} d\boldsymbol{z} d\boldsymbol{x}^{(v)} \\
&= \int \int p(\boldsymbol{x}^{(v)}|\boldsymbol{z}) p(\boldsymbol{x}^{(v)}) \log p(\boldsymbol{x}^{(v)}|\boldsymbol{z}) d\boldsymbol{z} d\boldsymbol{x}^{(v)} \\
&\quad + \int \int p(\boldsymbol{x}^{(v)}|\boldsymbol{z}) p(\boldsymbol{z}) \log \frac{1}{p(\boldsymbol{z})} d\boldsymbol{z} d\boldsymbol{x}^{(v)}.
\end{aligned}
$$

(23)

Since $H(\boldsymbol{z}) = - \int p(\boldsymbol{z}) \log p(\boldsymbol{z}) d\boldsymbol{z} \geq 0$, we have

$$
\begin{aligned}
I(\boldsymbol{x}^{(v)}; \boldsymbol{z}) &= \int \int p(\boldsymbol{x}^{(v)}|\boldsymbol{z}) p(\boldsymbol{x}^{(v)}) \log p(\boldsymbol{x}^{(v)}|\boldsymbol{z}) d\boldsymbol{z} d\boldsymbol{x}^{(v)} \\
&\quad + \int p(\boldsymbol{x}^{(v)}|\boldsymbol{z}) H(\boldsymbol{z}) d\boldsymbol{z} \\
&\geq \int \int p(\boldsymbol{x}^{(v)}|\boldsymbol{z}) p(\boldsymbol{x}^{(v)}) \log p(\boldsymbol{x}^{(v)}|\boldsymbol{z}) d\boldsymbol{z} d\boldsymbol{x}^{(v)} \\
&= \int \int p(\boldsymbol{x}^{(v)}|\boldsymbol{z}) p(\boldsymbol{x}^{(v)}) \log q^v(\boldsymbol{x}^{(v)}|\boldsymbol{z}) d\boldsymbol{z} d\boldsymbol{x}^{(v)} \\
&\quad + \int \int p(\boldsymbol{x}^{(v)}|\boldsymbol{z}) p(\boldsymbol{x}^{(v)}) \log \frac{p(\boldsymbol{x}^{(v)}|\boldsymbol{z})}{q^v(\boldsymbol{x}^{(v)}|\boldsymbol{z})} d\boldsymbol{z} d\boldsymbol{x}^{(v)}.
\end{aligned}
$$

(24)

Based on the definition of the Kullback-Leibler divergence, we can get

$$
D_{KL}(p(\boldsymbol{x}^{(v)}|\boldsymbol{z}) \| q^v(\boldsymbol{x}^{(v)}|\boldsymbol{z})) = \int p(\boldsymbol{x}^{(v)}|\boldsymbol{z}) \log \frac{p(\boldsymbol{x}^{(v)}|\boldsymbol{z})}{q^v(\boldsymbol{x}^{(v)}|\boldsymbol{z})} d\boldsymbol{x}^{(v)}.
$$

(25)

Since $D_{KL}(p(\boldsymbol{x}^{(v)}|\boldsymbol{z}) \| q^v(\boldsymbol{x}^{(v)}|\boldsymbol{z})) \geq 0$, we have

$$
\begin{aligned}
I(\boldsymbol{x}^{(v)}; \boldsymbol{z}) &\geq \int \int p(\boldsymbol{x}^{(v)}|\boldsymbol{z}) p(\boldsymbol{x}^{(v)}) \log q^v(\boldsymbol{x}^{(v)}|\boldsymbol{z}) d\boldsymbol{z} d\boldsymbol{x}^{(v)} \\
&\quad + \int p(\boldsymbol{x}^{(v)}) D_{KL}(p(\boldsymbol{x}^{(v)}|\boldsymbol{z}) \| q^v(\boldsymbol{x}^{(v)}|\boldsymbol{z})) d\boldsymbol{z} \\
&\geq \int \int p(\boldsymbol{x}^{(v)}|\boldsymbol{z}) p(\boldsymbol{x}^{(v)}) \log q^v(\boldsymbol{x}^v|\boldsymbol{z}) d\boldsymbol{z} d\boldsymbol{x}^{(v)} \\
&= \int \int p(\boldsymbol{z}, \boldsymbol{x}^{(v)}) \log q^v(\boldsymbol{x}^{(v)}|\boldsymbol{z}) d\boldsymbol{z} d\boldsymbol{x}^{(v)}.
\end{aligned}
$$

(26)

According to the property of integration, we have $\int \int p(\boldsymbol{z}, \boldsymbol{x}^{(v)}) \log q^v(\boldsymbol{x}^{(v)}|\boldsymbol{z}) d\boldsymbol{z} d\boldsymbol{x}^{(v)} = \int \int p(\{\boldsymbol{x}\}, \boldsymbol{z}) \log q^v(\boldsymbol{x}^{(v)}|\boldsymbol{z}) d\{\boldsymbol{x}\} d\boldsymbol{z}$, Thus, Eq. (26) can be rewritten as

$$
\begin{aligned}
I&(\boldsymbol{x}^{(v)}; \boldsymbol{z}) \\
&\geq \int \int p(\{\boldsymbol{x}\}, \boldsymbol{z}) \log q^v(\boldsymbol{x}^{(v)}|\boldsymbol{z}) d\{\boldsymbol{x}\} d\boldsymbol{z} \\
&= \int p(\{\boldsymbol{x}\}) \int p(\boldsymbol{z}|\{\boldsymbol{x}\}) \log q^v(\boldsymbol{x}^{(v)}|\boldsymbol{z}) d\{\boldsymbol{x}\} d\boldsymbol{z} \\
&= \mathbb{E}_{\boldsymbol{x} \sim p(\{\boldsymbol{x}\})} \left[ \int p(\boldsymbol{z}|\{\boldsymbol{x}\}) \log q^v(\boldsymbol{x}^{(v)}|\boldsymbol{z}) d\boldsymbol{z} \right]
\end{aligned}
\tag{27}
$$

Regarding the second term $I(\{\boldsymbol{x}^{\sim(v)}\}; \boldsymbol{z} \mid \boldsymbol{x}^{(v)})$, we have the following expansion based on the definition of conditional mutual information $I(a; b \mid c) = \iiint p(a, b, c) \log \left( \frac{p(a,b|c)}{p(a|c)p(b|c)} \right) da\,db\,dc$:

$$
\begin{aligned}
I&(\{\boldsymbol{x}^{\sim(v)}\}; \boldsymbol{z}|\boldsymbol{x}^{(v)}) \\
&= \int \int p(\{\boldsymbol{x}\}, \boldsymbol{z}) \log \frac{p(\{\boldsymbol{x}\}, \boldsymbol{z})p(\boldsymbol{x}^{(v)})}{p(\{\boldsymbol{x}\})p(\boldsymbol{z}, \boldsymbol{x}^{(v)})} d\{\boldsymbol{x}\} d\boldsymbol{z} \\
&= \int \int p(\{\boldsymbol{x}\}, \boldsymbol{z}) \log \frac{p(\boldsymbol{z}|\{\boldsymbol{x}\})}{p(\boldsymbol{z}|\boldsymbol{x}^{(v)})} d\{\boldsymbol{x}\} d\boldsymbol{z}
\end{aligned}
\tag{28}
$$

By introducing the variational distribution $g^v(\boldsymbol{z}|\boldsymbol{x}^{(v)})$ to estimate $p(\boldsymbol{z}|\boldsymbol{x}^{(v)})$, we have

$$
\begin{aligned}
I&(\{\boldsymbol{x}^{\sim(v)}\}; \boldsymbol{z}|\boldsymbol{x}^{(v)}) \\
&= \int \int p(\{\boldsymbol{x}\}, \boldsymbol{z}) \log \frac{p(\boldsymbol{z}|\{\boldsymbol{x}\})}{g^v(\boldsymbol{z}|\boldsymbol{x}^{(v)})} d\{\boldsymbol{x}\} d\boldsymbol{z} + \int \int p(\{\boldsymbol{x}\}, \boldsymbol{z}) \log \frac{g^v(\boldsymbol{z}|\boldsymbol{x}^{(v)})}{p(\boldsymbol{z}|\boldsymbol{x}^{(v)})} d\{\boldsymbol{x}\} d\boldsymbol{z} \\
&= \int \int p(\{\boldsymbol{x}\}, \boldsymbol{z}) \log \frac{p(\boldsymbol{z}|\{\boldsymbol{x}\})}{g^v(\boldsymbol{z}|\boldsymbol{x}^{(v)})} d\{\boldsymbol{x}\} d\boldsymbol{z} + \int p(\{\boldsymbol{x}\}) \int p(\boldsymbol{z}|\{\boldsymbol{x}\}) \log \frac{g^v(\boldsymbol{z}|\boldsymbol{x}^{(v)})}{p(\boldsymbol{z}|\boldsymbol{x}^{(v)})} d\{\boldsymbol{x}\} d\boldsymbol{z} \\
&= \int \int p(\{\boldsymbol{x}\}, \boldsymbol{z}) \log \frac{p(\boldsymbol{z}|\{\boldsymbol{x}\})}{g^v(\boldsymbol{z}|\boldsymbol{x}^{(v)})} d\{\boldsymbol{x}\} d\boldsymbol{z} + \int p(\boldsymbol{x}^{(v)}) \int p(\boldsymbol{z}|\boldsymbol{x}^{(v)}) \log \frac{g^v(\boldsymbol{z}|\boldsymbol{x}^{(v)})}{p(\boldsymbol{z}|\boldsymbol{x}^{(v)})} d\boldsymbol{x}^{(v)} d\boldsymbol{z} \\
&= \int \int p(\{\boldsymbol{x}\}, \boldsymbol{z}) \log \frac{p(\boldsymbol{z}|\{\boldsymbol{x}\})}{g^v(\boldsymbol{z}|\boldsymbol{x}^{(v)})} d\{\boldsymbol{x}\} d\boldsymbol{z} - \int p(\boldsymbol{x}^{(v)}) D_{KL}(p(\boldsymbol{z}|\boldsymbol{x}^{(v)})||g^v(\boldsymbol{z}|\boldsymbol{x}^{(v)})) d\boldsymbol{x}^{(v)} \\
&\leq \int \int p(\{\boldsymbol{x}\}, \boldsymbol{z}) \log \frac{p(\boldsymbol{z}|\{\boldsymbol{x}\})}{g^v(\boldsymbol{z}|\boldsymbol{x}^{(v)})} d\{\boldsymbol{x}\} d\boldsymbol{z} \\
&= \mathbb{E}_{\{\boldsymbol{x}\} \sim p(\{\boldsymbol{x}\})} \left[ D_{KL} \left( p(\boldsymbol{z}|\{\boldsymbol{x}\})||g^v(\boldsymbol{z}|\boldsymbol{x}^{(v)}) \right) \right],
\end{aligned}
\tag{29}
$$

where $p(\boldsymbol{z}|\{\boldsymbol{x}\})$ is determined by the distribution fusion method provided in Eq. 14

Regarding the category-level constraint $\min I(\boldsymbol{x}^{(v)}; \boldsymbol{y}) - I(\boldsymbol{z}^{(v)}; \boldsymbol{y})$ in the of model (4), we have the following equivalent transformation according to the equality $I(\boldsymbol{x}^{(v)}; \boldsymbol{y}) = H(\boldsymbol{y}) - H(\boldsymbol{y}|\boldsymbol{x}^{(v)})$

$$
\min I(\boldsymbol{x}^{(v)}; \boldsymbol{y}) - I(\boldsymbol{z}^{(v)}; \boldsymbol{y}) \iff \min H(\boldsymbol{y}|\boldsymbol{z}^{(v)}) - H(\boldsymbol{y}|\boldsymbol{x}^{(v)}).
\tag{30}
$$

Utilizing the definition of entropy, we can get

$$
H(\boldsymbol{y} \mid \boldsymbol{z}^{(v)}) = - \int p(\boldsymbol{y} \mid \boldsymbol{z}^{(v)}) \log p(\boldsymbol{y} \mid \boldsymbol{z}^{(v)}) d\boldsymbol{y}.
\tag{31}
$$

Thus, we can find that $H(\boldsymbol{y} \mid \boldsymbol{z}^{(v)})$ depends solely on $p(\boldsymbol{y} \mid \boldsymbol{z}^{(v)})$. As a result, minimizing $I(\boldsymbol{x}^{(v)}; \boldsymbol{y}) - I(\boldsymbol{z}^{(v)}; \boldsymbol{y})$ naturally transforms into optimizing the distance between the corresponding two distributions:

$$
\min \sum_{v \in \mathcal{V}} D_{KL} \left( p(\boldsymbol{y}|\boldsymbol{z}^{(v)})||p(\boldsymbol{y}|\boldsymbol{x}^{(v)}) \right).
\tag{32}
$$

For the last term $I(\boldsymbol{y}; \boldsymbol{z}^{(v)} \mid \{\boldsymbol{z}^{\sim(v)}\})$, we have

$$
\begin{aligned}
& I(\boldsymbol{y}; \boldsymbol{z}_i | \{\boldsymbol{z}^{\sim(v)}\}) \\
=& H(\boldsymbol{y}|\{\boldsymbol{z}^{\sim(v)}\}) - H(\boldsymbol{y}|\{\boldsymbol{z}\}) \\
=& -\int p(\boldsymbol{y}|\{\boldsymbol{z}^{\sim(v)}\}) \log p(\boldsymbol{y}|\{\boldsymbol{z}^{\sim(v)}\}) d\boldsymbol{y} + \int p(\boldsymbol{y}|\{\boldsymbol{z}\}) \log p(\boldsymbol{y}|\{\boldsymbol{z}\}) d\boldsymbol{y} \\
=& -\int p(\boldsymbol{y}|\{\boldsymbol{z}^{\sim(v)}\}) \log \left[ \frac{p(\boldsymbol{y}|\{\boldsymbol{z}^{\sim(v)}\})}{p(\boldsymbol{y}|\{\boldsymbol{z}\})} p(\boldsymbol{y}|\{\boldsymbol{z}\}) \right] d\boldsymbol{y} \\
& + \int p(\boldsymbol{y}|\{\boldsymbol{z}\}) \log \left[ \frac{p(\boldsymbol{y}|\{\boldsymbol{z}\})}{p(\boldsymbol{y}|\{\boldsymbol{z}^{\sim(v)}\})} p(\boldsymbol{y}|\{\boldsymbol{z}^{\sim(v)}\}) \right] d\boldsymbol{y},
\end{aligned}
\tag{33}
$$

By adding terms in the logarithmic operation, we can obtain:

$$
\begin{aligned}
& I(\boldsymbol{y}; \boldsymbol{z}_i | \{\boldsymbol{z}^{\sim(v)}\}) \\
=& -\int p(\boldsymbol{y}|\{\boldsymbol{z}^{\sim(v)}\}) \log \left[ \frac{p(\boldsymbol{y}|\{\boldsymbol{z}^{\sim(v)}\})}{p(\boldsymbol{y}|\{\boldsymbol{z}\})} p(\boldsymbol{y}|\{\boldsymbol{z}\}) \right] d\boldsymbol{y} \\
& + \int p(\boldsymbol{y}|\{\boldsymbol{z}\}) \log \left[ \frac{p(\boldsymbol{y}|\{\boldsymbol{z}\})}{p(\boldsymbol{y}|\{\boldsymbol{z}^{\sim(v)}\})} p(\boldsymbol{y}|\{\boldsymbol{z}^{\sim(v)}\}) \right] d\boldsymbol{y} \\
=& -\int p(\boldsymbol{y}|\{\boldsymbol{z}^{\sim(v)}\}) \log \left[ \frac{p(\boldsymbol{y}|\{\boldsymbol{z}^{\sim(v)}\})}{p(\boldsymbol{y}|\{\boldsymbol{z}\})} \right] d\boldsymbol{y} - \int p(\boldsymbol{y}|\{\boldsymbol{z}^{\sim(v)}\}) \log p(\boldsymbol{y}|\{\boldsymbol{z}\}) d\boldsymbol{y} \\
& + \int p(\boldsymbol{y}|\{\boldsymbol{z}\}) \log \left[ \frac{p(\boldsymbol{y}|\{\boldsymbol{z}\})}{p(\boldsymbol{y}|\{\boldsymbol{z}^{\sim(v)}\})} \right] d\boldsymbol{y} + \int p(\boldsymbol{y}|\{\boldsymbol{z}\}) \log p(\boldsymbol{y}|\{\boldsymbol{z}^{\sim(v)}\}) d\boldsymbol{y}. \\
=& -D_{KL}(p(\boldsymbol{y}|\{\boldsymbol{z}^{\sim(v)}\})||p(\boldsymbol{y}|\{\boldsymbol{z}\})) + H(p(\boldsymbol{y}|\{\boldsymbol{z}^{\sim(v)}\}); p(\boldsymbol{y}|\{\boldsymbol{z}\})) \\
& + D_{KL}(p(\boldsymbol{y}|\{\boldsymbol{z}\})||p(\boldsymbol{y}|\{\boldsymbol{z}^{\sim(v)}\})) - H(p(\boldsymbol{y}|\{\boldsymbol{z}\}); p(\boldsymbol{y}|\{\boldsymbol{z}^{\sim(v)}\})) \\
\leq& D_{KL}(p(\boldsymbol{y}|\{\boldsymbol{z}\})||p(\boldsymbol{y}|\{\boldsymbol{z}^{\sim(v)}\})) + H(p(\boldsymbol{y}|\{\boldsymbol{z}^{\sim(v)}\}); p(\boldsymbol{y}|\{\boldsymbol{z}\})).
\end{aligned}
\tag{34}
$$

Since the optimization goals of $D_{KL}(p(\boldsymbol{y}|\{\boldsymbol{z}\})||p(\boldsymbol{y}|\{\boldsymbol{z}^{\sim(v)}\}))$ and $H(p(\boldsymbol{y}|\{\boldsymbol{z}^{\sim(v)}\}); p(\boldsymbol{y}|\{\boldsymbol{z}\}))$ are both to make the distributions of $p(\boldsymbol{y}|\{\boldsymbol{z}\})$ and $p(\boldsymbol{y}|\{\boldsymbol{z}^{\sim(v)}\}))$ closer, we adopt the tractable term $D_{KL}(p(\boldsymbol{y}|\{\boldsymbol{z}\})||p(\boldsymbol{y}|\{\boldsymbol{z}^{\sim(v)}\}))$ to minimize $I(\boldsymbol{y}; \boldsymbol{z}^{(v)} \mid \{\boldsymbol{z}^{\sim(v)}\})$. For the term $I(\boldsymbol{y}; \boldsymbol{z}^{(v)})$ in model (4), its purpose is to enhance the information correlation between the representation and the labels, which can be achieved by optimizing the classification loss. In conclusion, the final derived training loss is as follows:

$$
\begin{cases}
\mathcal{L}_f = \dfrac{1}{|\mathcal{V}|} \sum_{v \in \mathcal{V}} \left[ -\mathbb{E}_{\boldsymbol{z} \sim p(\boldsymbol{z}|\{\boldsymbol{x}\})} \log q^v(\boldsymbol{x}^{(v)}|\boldsymbol{z}) + D_{KL}\left( p(\boldsymbol{z}|\{\boldsymbol{x}\})||g^v(\boldsymbol{z}|\boldsymbol{x}^{(v)}) \right) \right] \\
\mathcal{L}_c = \dfrac{1}{|\mathcal{V}|} \sum_{v \in \mathcal{V}} \left( D_{KL}\left( p(\boldsymbol{y}|\boldsymbol{z}^{(v)})||p(\boldsymbol{y}|\boldsymbol{x}^{(v)}) \right) + D_{KL}\left( p(\boldsymbol{y}|\{\boldsymbol{z}\}) \| p(\boldsymbol{y}|\{\boldsymbol{z}^{\sim(v)}\}) \right) \right).
\end{cases}
\tag{35}
$$

## A.3 TRAINING PROCESS

The main training process of SMVTEP is summarised in Algorithm 1.

## A.4 EXPERIMENT

### A.4.1 EXPERIMENT SETUP

**Datasets and Comparison Methods.** We conduct experiments on six publicly available multi-view multi-label datasets, summarized in Table 3. The details of these datasets are as follows. **Corel 5k** contains 4999 images with 260 annotation words, where each word can be treated as a semantic label. **IAPRTC12** consists of 19,627 high-resolution natural images with 261 possible labels, covering categories such as sports, animals, human activities, and urban scenes. **ESPGame** provides 20,770 images with 268 associated tags collected from an online image labeling game. **Pascal07**

---

**Algorithm 1** Training process of TITRL

---

**Input:** Incomplete multi-view multi-label data $(\{\boldsymbol{x}^{(v)}\}_{v=1}^V, \boldsymbol{y})$, observed view set $\mathcal{V}$, known label set $\mathcal{U}$, and balanced parameters $\lambda_1$ and $\lambda_2$.
**Output:** Prediction $\boldsymbol{p}^t$.
 1: Initialize the parameters $\{\boldsymbol{b_i}\}_{i=1}^c$ as $\boldsymbol{I} \in \{0,1\}^{c \times c}$
 2: **for** t = 0; t < Total epoch; t ++ **do**
 3:    Compute the variational distribution $\{g^v(\boldsymbol{z} \mid \boldsymbol{x}^{(v)}) \mid v \in \mathcal{V}\}$ of the shared representations extracted from each available view through encoders $f_\mu^v$ and $f_{\sigma^2}^v$.
 4:    Compute the distribution parameters of the integrated shared representation $\boldsymbol{z}$ by Eq. (14)
 5:    Sample $\boldsymbol{z}$ from the corresponding distribution by Eq. (15).
 6:    Compute the conditional distribution $\{q^v(\mathbf{x}^{(v)} \mid \mathbf{z}) \mid v \in \mathcal{V}\}$ by decoders
 7:    Obtain the distribution of $c$ label prototype $\{\boldsymbol{l}_i \sim \mathcal{N}(h_\mu(\boldsymbol{b}_i), h_{\sigma^2}(\boldsymbol{b}_i)\boldsymbol{I}) \mid i = 1, ..., c\}$.
 8:    Sample each $\boldsymbol{l}_i$ from the corresponding distribution like Eq. (15).
 9:    Compute the pseudo-predictions $p(\boldsymbol{y}_i \mid \{\boldsymbol{z}^{\sim(v)}\})$, $p(\boldsymbol{y}_i \mid \boldsymbol{x}^{(v)})$, and $p(\boldsymbol{y}_i \mid \boldsymbol{z}^{(v)})$ by Eq. (15).
10:    Compute the final prediction $\boldsymbol{p}^t$ by Eq. (18)
11:    Compute the total loss $\mathcal{L}$ by Eq. (20).
12:    Update network parameters.
13: **end for**

---

is a widely used benchmark for object detection and recognition, comprising 9963 images across 20 object categories. **Mirflickr** includes 25,000 Flickr images, each annotated with up to 38 labels. **OBJECT** contains 6047 samples described from five different perspectives and annotated with 31 attributes. To demonstrate the effectiveness of our method, we compare it with nine representative approaches: AIMNet Liu et al. (2024a), DICNet Liu et al. (2023b), DIMC Wen et al. (2023), iMVWL Tan et al. (2018), LMVCAT Liu et al. (2023c), MTD Liu et al. (2023a), SIP Liu et al. (2024b), LVSL Zhao et al. (2022), and DM2L Ma & Chen (2021). A detailed summary of these comparison methods, including their sources and functionality, is presented in Table 4.

Table 3: Detailed information of datasets.

| View | Object | VOC 2007 | Corel 5k | ESP Game | IAPR TC-12 | MIR Flickr |
|---|---|---|---|---|---|---|
| 1 | CH (64) | DenseHue (100) | DenseHue (100) | DenseHue (100) | DenseHue (100) | DenseHue (100) |
| 2 | CM (225) | DenseSift (1000) | DenseSift (1000) | DenseSift (1000) | DenseSift (1000) | DenseSift (1000) |
| 3 | CORR (144) | GIST (512) | GIST (512) | GIST (512) | GIST (512) | GIST (512) |
| 4 | EDH (73) | HSV (4096) | HSV (4096) | HSV (4096) | HSV (4096) | HSV (4096) |
| 5 | WT (128) | RGB (4096) | RGB (4096) | RGB (4096) | RGB (4096) | RGB (4096) |
| 6 | – | LAB (4096) | LAB (4096) | LAB (4096) | LAB (4096) | LAB (4096) |
| #Labels | 31 | 20 | 260 | 268 | 291 | 38 |
| #Instances | 6047 | 9963 | 4999 | 20770 | 19627 | 25000 |

Table 4: Detailed information of comparison methods. ✓ indicates the method can handle the corresponding problem, while ✗ denotes it cannot.

| Method | Source | Year | Multi-label | Multi-view | Missing-view | Missing-label |
|---|---|---|---|---|---|---|
| iMVWL | IJCAI | 2018 | ✓ | ✓ | ✓ | ✓ |
| DM2L | PR | 2021 | ✓ | ✗ | ✗ | ✓ |
| LVLS | TMM | 2022 | ✓ | ✓ | ✗ | ✗ |
| DICNet | AAAI | 2023 | ✓ | ✓ | ✓ | ✓ |
| DIMC | TNNLS | 2023 | ✓ | ✓ | ✓ | ✓ |
| LMVCAT | AAAI | 2023 | ✓ | ✓ | ✓ | ✓ |
| AIMNet | AAAI | 2024 | ✓ | ✓ | ✓ | ✓ |
| MTD | NeurIPS | 2024 | ✓ | ✓ | ✓ | ✓ |
| SIP | ICML | 2024 | ✓ | ✓ | ✓ | ✓ |

**Implementation Details.** We adopt six widely used evaluation metrics to ensure consistency across experiments: Hamming Loss (HL), Ranking Loss (RL), OneError (OE), Coverage (Cov), Average

Precision (AP), and Area Under Curve (AUC). In general, higher values of AP and AUC indicate stronger predictive ability, while lower HL, RL, OE, and Cov reflect better classification accuracy. Specifically: (1) HL measures the proportion of incorrectly predicted labels; (2) RL quantifies the proportion of label pairs that are incorrectly ordered; (3) OE checks whether the top-ranked predicted label matches the ground truth; (4) Cov evaluates how many predicted labels need to be considered to cover all true labels; (5) AP corresponds to the average precision across different recall levels; and (6) AUC represents the probability that a randomly chosen positive instance is ranked above a randomly chosen negative one. For all datasets, the neighbor number $k$ is set to 10. Optimization is performed using the Adam optimizer with an initial learning rate of 0.0001. All models share the same dataset partitioning, and the positions of missing views and labels are fixed across methods to guarantee fairness in comparison.

### A.4.2 EXPERIMENT RESULT

**Comparative Experiment.**

We provide the complete results on six datasets by varying PER and LMR from 30% to 90%. Fig. 4 reports the case where PER changes while LMR is fixed at 50%. As expected, performance gradually decreases for all methods as more views are removed, but TITRL consistently secures the best accuracy. The performance gap over baselines becomes more significant at higher missing levels, indicating that TITRL is particularly effective at mitigating the negative impact of severe view absence through robust multi-view representation learning. Fig. 5 presents the results of varying LMR with PER fixed at 50%. Compared with the PER case, the decline under missing labels is generally smoother, yet TITRL still shows clear advantages across all datasets, which confirms the effectiveness of the label semantic learning in our method. Thus, TITRL maintains stable and robust performance under both feature absence and label incompleteness, highlighting its applicability to challenging real-world scenarios.

To provide a more comprehensive multi-metric visualization of performance, we further present radar plots under various conditions of data sparsity. Fig. 8 illustrates the results with a fixed LMR of 50% while PER increases from 50% to 90%. It is evident that TITRL consistently occupies the outermost boundary across all datasets and PER levels. This visually confirms its superior performance across all six evaluation metrics. Even as the view missingness becomes more severe (e.g., PER=90%), TITRL maintains a significant performance margin over the competing methods, highlighting its robustness against feature absence.

Furthermore, Fig. 9 and Fig. 10 depict scenarios with even higher label missingness (LMR=70% and LMR=90%, respectively). In these highly challenging settings, TITRL's superiority remains unshaken. As data incompleteness intensifies, the performance of many baseline methods degrades substantially, whereas TITRL sustains its strong performance profile. This demonstrates the exceptional resilience of our method in handling extreme levels of label sparsity. Collectively, these radar plots offer strong visual evidence that TITRL not only achieves the highest performance but also maintains remarkable robustness, solidifying its effectiveness for real-world iMvMLC tasks.

**Parameter Sensitivity.**

To investigate the sensitivity of the model's hyperparameters, we perform a grid search for parameter selection on $\lambda_1$ and $\lambda_2$, and report the validation AP on Corel5k, ESPGame, and IAPRTC12. The results are visualized as heatmaps in Fig. 6. The values of both parameters are adjusted within the range $\{0.001, 0.005, 0.01, 0.05, 0.1, 0.5, 1\}$. It can be observed that TITRL is more responsive to $\lambda_1$, as an improper selection of $\lambda_1$ can lead to a significant performance degradation. The optimal configuration tends to appear when $\lambda_1$ lies within the range $(0.01, 0.05)$ while $\lambda_2$ approaches 1, a trend that holds consistently across both small-scale datasets (e.g., Corel 5k, IAPRTC12) and large-scale datasets (e.g., Mirflickr, OBJECT). These observations suggest that TITRL requires careful tuning of $\lambda_1$ to promote the semantic compactness of the extracted representations, whereas $\lambda_2$ can be set to a relatively high value to stabilize performance. Overall, TITRL maintains stable accuracy across a wide parameter range, with clearly identifiable regions of optimality.

**Convergence Behavior.**

We further examine the optimization behavior of TITRL by tracking training and validation dynamics over epochs. Fig. 7 presents the evolution curves of training loss and validation AP on Corel5k,

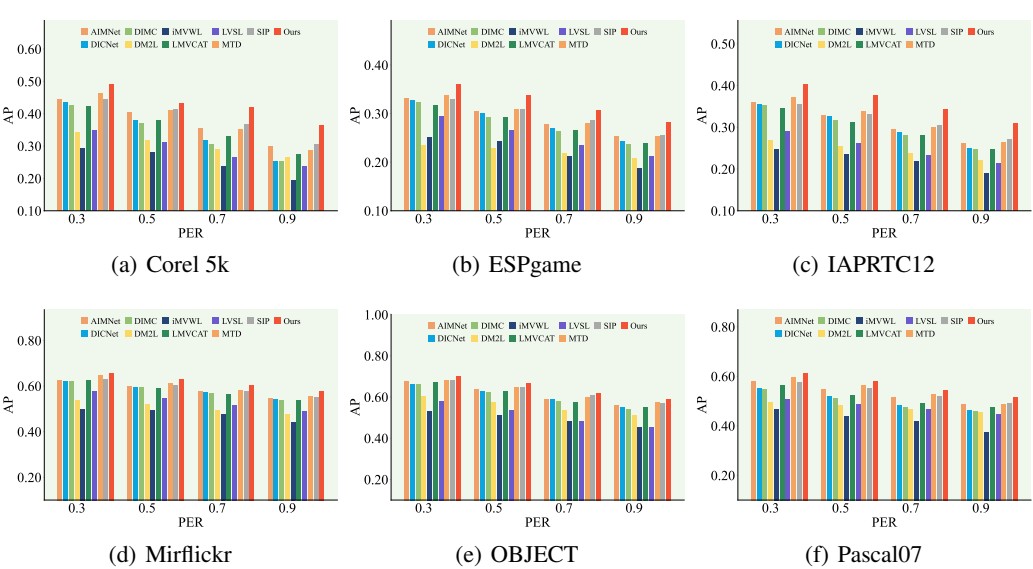

Figure 4: Results on six datasets with PER changing from 30% to 90%, and LMR fixed at 50%

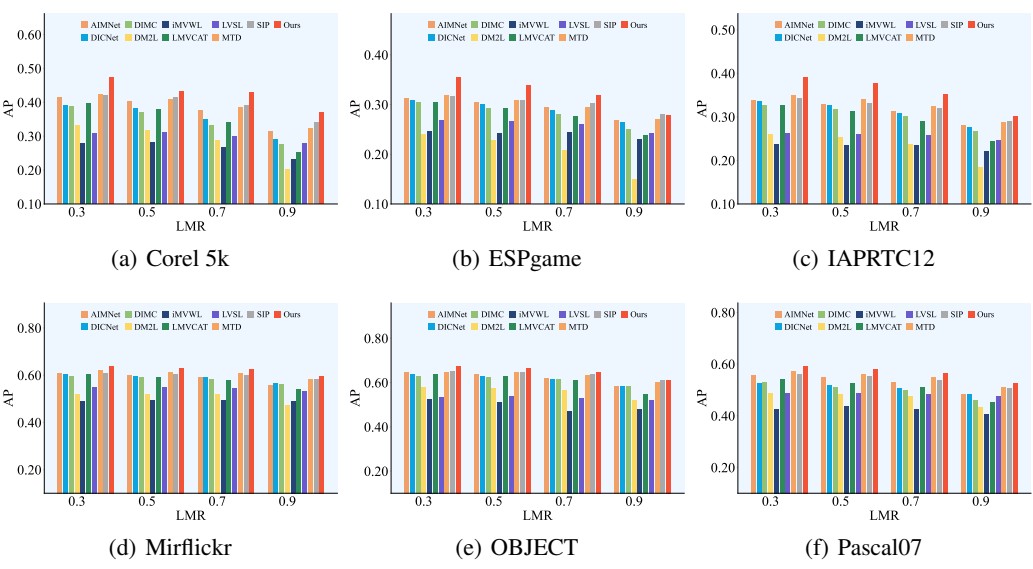

Figure 5: Results on six datasets with LMR changing from 30% to 90%, and PER fixed at 50%

ESPGame, and IAPRTC12. From the result, we can find that the training loss decreases smoothly, while validation AP rises quickly in the early epochs and stabilizes thereafter, typically converging within 25-40 epochs. This rapid and stable convergence demonstrates the efficiency of our optimization strategy and the well-conditioned nature of the proposed objective. Moreover, the close alignment between training and validation curves indicates that overfitting is effectively controlled. We also observe consistent convergence patterns across datasets and repeated runs, which confirms the numerical stability of TITRL under diverse conditions. These properties collectively highlight that TITRL not only achieves strong predictive accuracy under missing data but also provides reliable training dynamics, ensuring efficiency and reproducibility in practice.

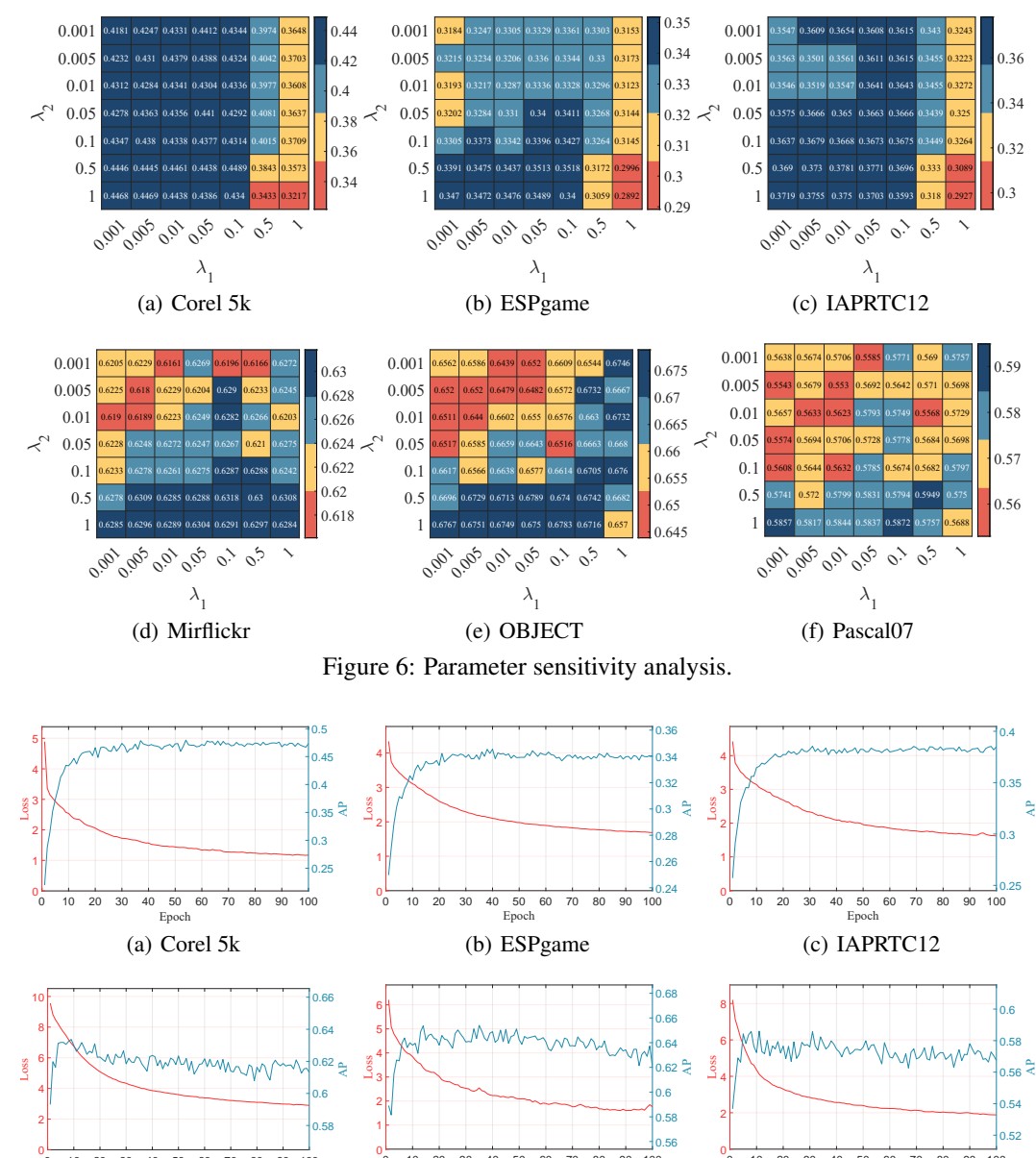

Figure 6: Parameter sensitivity analysis.

Figure 7: Convergence analysis.

## USE OF LARGE LANGUAGE MODELS (LLMS)

In the course of this research, Large Language Models (LLMs) were employed as an assistive tool for light English editing, such as grammar corrections, wording improvements, and enhancing clarity. The LLM played no role in the ideation, experimental design, data processing, analysis, or methodological development of the research. All technical content, including research ideas, methodology, data analysis, and results, were independently developed and verified by the authors, who take full responsibility for the manuscript.

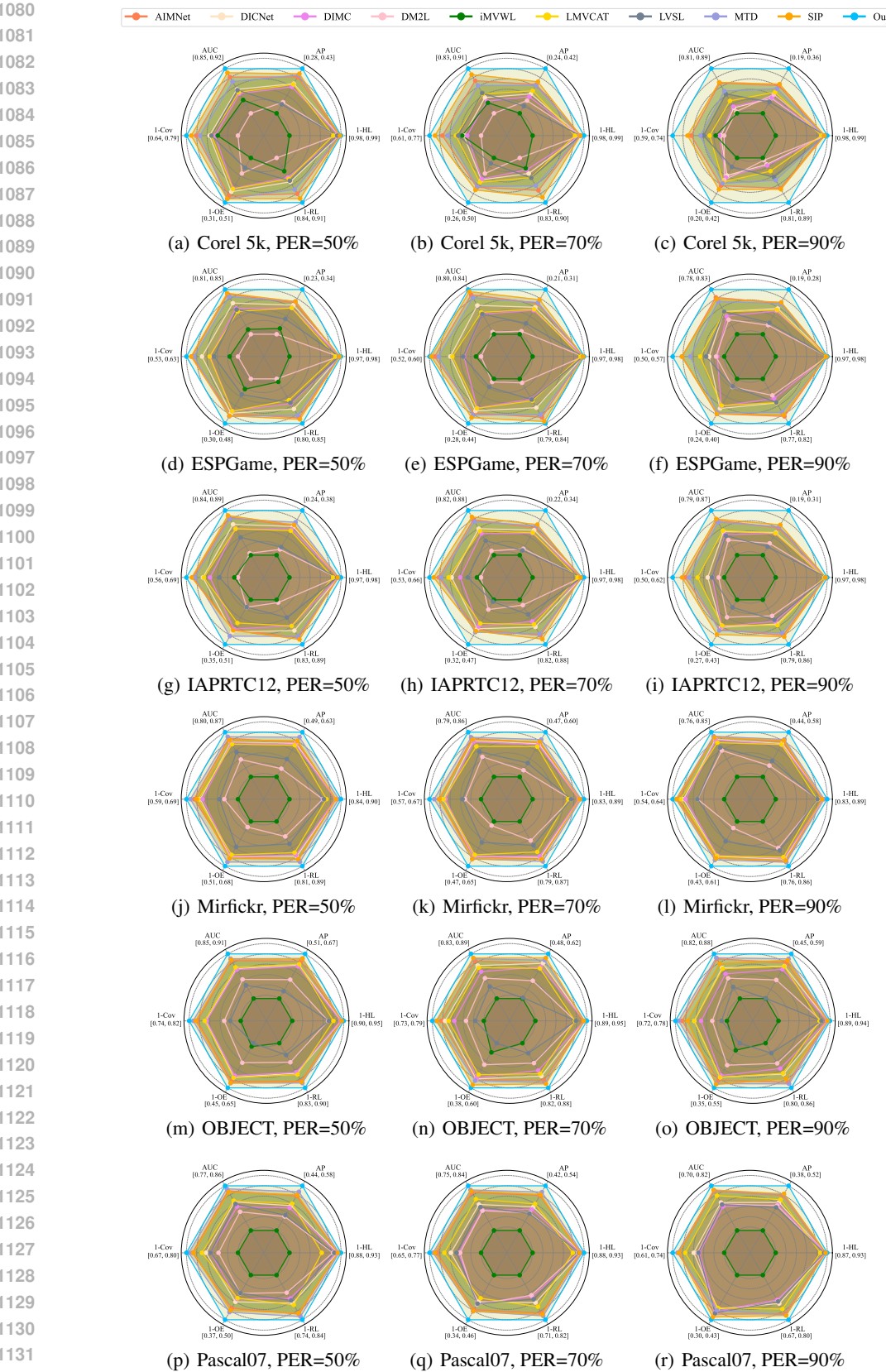

Figure 8: Experimental results of ten methods on six datasets with PER varying from 50% to 90%, while LMR= 50%.

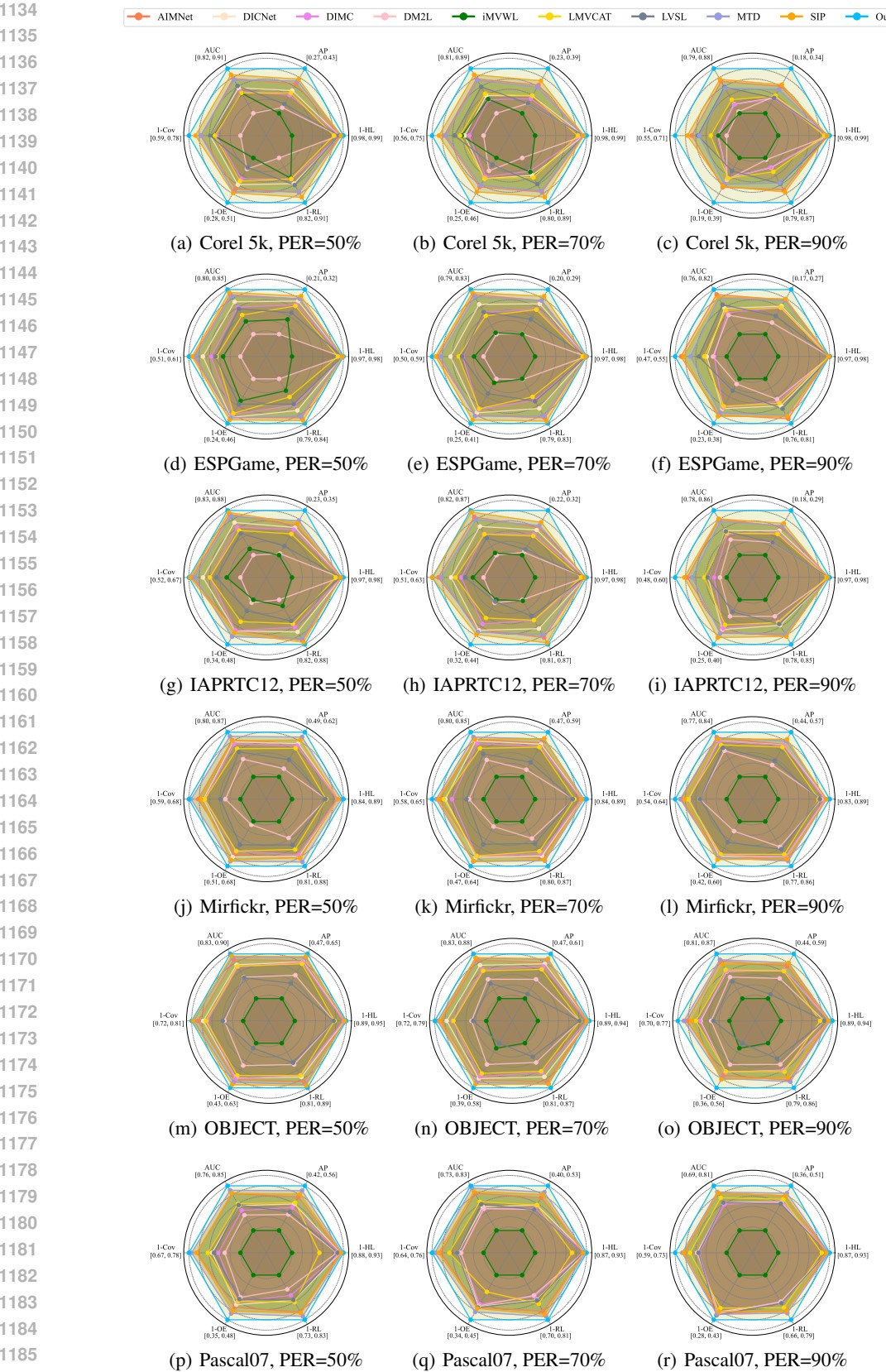

Figure 9: Experimental results of ten methods on six datasets with PER varying from 50% to 90%, while LMR= 70%.

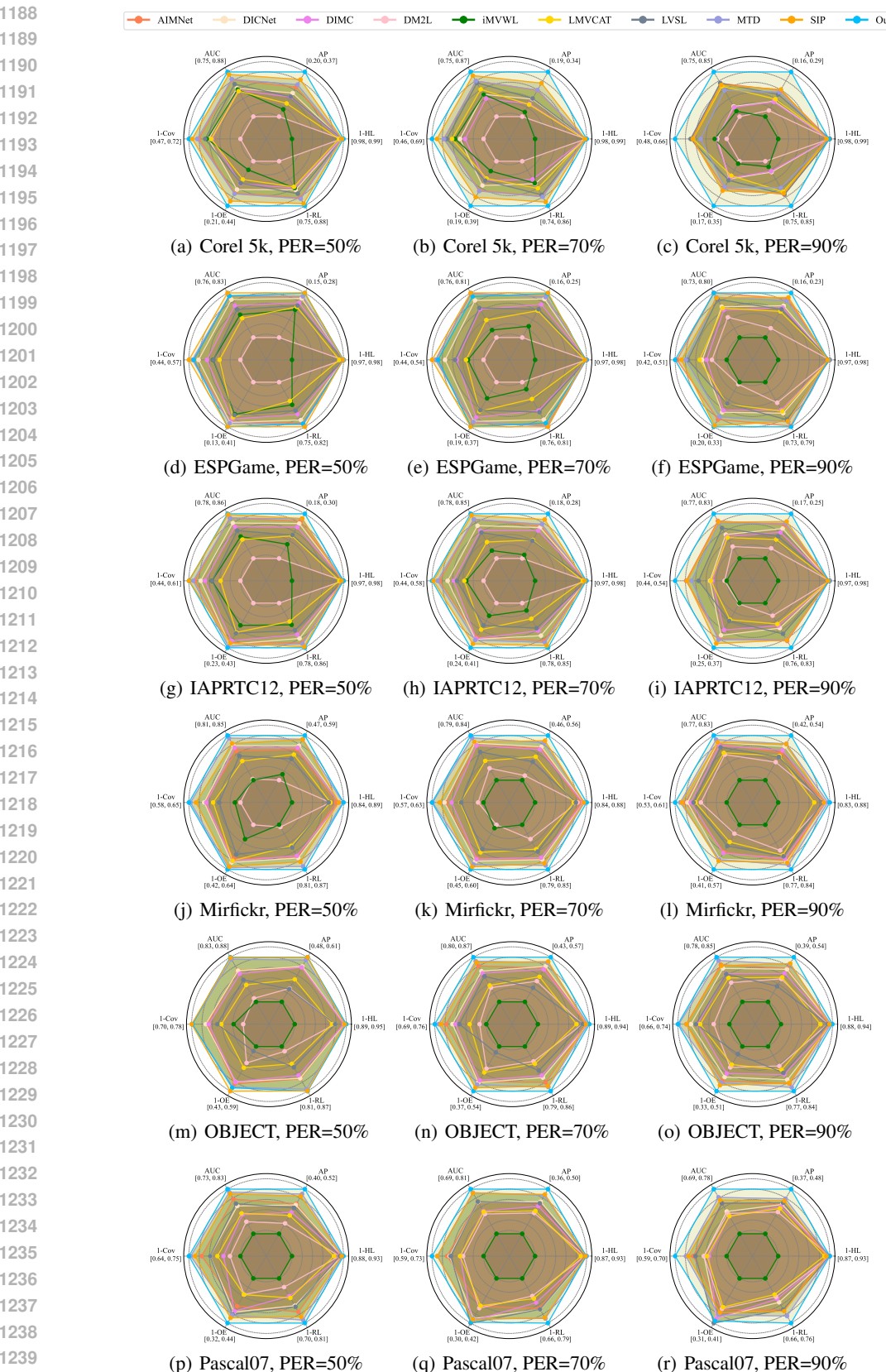

Figure 10: Experimental results of ten methods on six datasets with PER varying from 50% to 90%, while LMR= 90%.

