# OpenReview forum: "Theory-Inspired Task-Relevant Representation Learning for Incomplete Multi-View Multi-Label Learning"
_ICLR.cc/2026/Conference — ICLR 2026 Conference Withdrawn Submission_

### Official Review · Reviewer_9jZM · 2025-10-17

**Soundness:** 3
**Presentation:** 3
**Contribution:** 2
**Rating:** 4
**Confidence:** 4

**Summary:**

This paper also addresses the incomplete multi-view multi-label problem by proposing a theory-driven method named TITRL. Its core is a "Dual-Layer Constraint Framework" that applies constraints to representation learning at both the feature and category levels from an information-theoretic perspective. The goal is to learn task-relevant shared representations while actively suppressing view-specific noise. Additionally, the method combines an early, distribution-based view fusion with a late, confidence-based fusion strategy, and it introduces sample-level label correlation learning.

**Strengths:**

The paper's arguments are relatively rigorous. The framework diagram and the experimental section are clear, contributing to good readability.

**Weaknesses:**

Please see Questions.

**Questions:**

1)In the abstract, the paper mentions deriving "tractable bounds" for the mutual information model. Is this the first work to propose this concept? Also, the title includes "task-relevant." How is this concept specifically embodied in the method?

2)Regarding the definition of important symbols, the paper frequently uses the tilde. For example, x^{(v)} with its (~), does denote the set of all views other than the view v?

3)Why can the shared information term I_v^s not be optimized directly? The justification for the derivation from Equation 3 to the objective in Equation 4 is not entirely clear. Could you provide a more detailed explanation? Furthermore, since Equation 4 primarily constrains the shared component z, how does the method ensure that unique, view-specific information is also effectively integrated? Is the 'z' that appears in L_f computed according to Equation 15?

4)The rationale for using the Product-of-Experts (PoE) framework is not explained in detail. Could you elaborate on this choice, especially since other methods exist for capturing multi-dimensional constraints or complementary information in data?

5)Why does Equation 16 compute the similarity between a shared feature and a label prototype, which seem to exist in different semantic spaces? The initial label embeddings are not binary, and the stochastic encoder might introduce noise; how does the paper account for this? Additionally, the late fusion mechanism uses the model's own prediction confidence, L_con, as weights. Early in training, predictions are close to 0.5, leading to low and uniform confidence weights. Does this affect the stability and efficiency of convergence during the initial training phase?

6)The paper does not seem to explicitly mark the positions of present versus missing data. How does the model achieve good performance in this mixed-data scenario? For instance, regarding the category-level constraint L_c, specifically the term D_kl{p(y|z(v))||p(y|x(v))}. If the source view x(v) itself is noisy or contains spurious correlations with the labels, does this constraint risk forcing the "purer" latent representation z(v) to mimic this noisy predictive distribution, thereby contaminating the representation?

---

### Official Review · Reviewer_ndiy · 2025-10-31

**Soundness:** 3
**Presentation:** 3
**Contribution:** 2
**Rating:** 2
**Confidence:** 3

**Summary:**

This addresses a critical challenge in multi-view multi-label learning handling dual data incompleteness, where both feature views and labels are partially missing. The authors propose a method, that grounded in information theory, to extract task-relevant representations while mitigating view-specific noise and enhancing label semantic learning. The framework employs a dual-layer constraint mechanism at both feature and category levels, guiding the model toward purer shared representations and better label correlations. The proposed method incorporates a distribution-aware fusion strategy for early integration and confidence-driven late fusion, ensuring semantic stability and dynamic view quality assessment. Extensive experiments on six public datasets demonstrate the method's effectiveness over several baselines.

**Strengths:**

1. The dual issues of incomplete views and labels are well-motivated, as they reflect real-world scenarios in multi-modal learning tasks, such as remote sensing and medical imaging.
2. The use of mutual information as a guiding principle for representation learning is grounded in established theoretical concepts, providing a structured basis for the proposed method.
3. The paper evaluates its method across multiple datasets and metrics, providing a reasonably broad empirical foundation for assessing the approach's performance.
4. The paper is well written and easy to read.

**Weaknesses:**

1. While the variational bounds for mutual information are derived rigorously, the reliance on these approximations might introduce gaps between theoretical objectives and practical outcomes. It is unclear how sensitive the method is to the quality of these approximations, especially in highly noisy or sparse data scenarios.

2. The method emphasizes suppressing view-specific noise and optimizing shared information. However, in real-world applications, certain view-specific information could carry complementary insights rather than noise. The framework does not explicitly address how to balance suppressing noise while preserving useful view-specific details.

3. The ablation study evaluates key modules (dual-layer constraints, fusion strategies, and label correlation learning) but lacks finer granularity. For instance, it would be valuable to isolate the impact of specific components (e.g., early vs. late fusion) or analyze performance under extreme sparsity conditions.

4. The framework assumes that mutual information can be effectively optimized in high-dimensional multi-view data. However, the computational cost of variational inference and distribution-aware fusion might become prohibitive as the number of views or labels grows significantly.

**Questions:**

Please refer to the weaknesses above.

---

### Official Review · Reviewer_mVq9 · 2025-10-31

**Soundness:** 3
**Presentation:** 3
**Contribution:** 2
**Rating:** 2
**Confidence:** 4

**Summary:**

The paper tackles incomplete multi-view multi-label classification (iMvMLC), where both views and labels are partially missing, complicating shared representation learning, view fusion, and label semantics. TITRL introduces a dual-layer information-theoretic framework to extract task-relevant shared representations by maximizing mutual information (MI) with each view while suppressing view-specific redundancy, and aligning representation.

**Strengths:**

1.The framework grounds representation learning in dual-layer MI objectives with explicit variational bounds for optimization.
2. The use of stochastic label prototypes and instance-level alignment captures sample-specific label correlations beyond fixed pairwise graphs.

**Weaknesses:**

1. The diagonal-Gaussian posterior and PoE aggregation may be restrictive under multimodal or non-Gaussian view distributions and alternatives are not compared.
2. How does the author ensure that the extracted label prototype represents the semantics? Are there any experiments or visualization results that can support this claim? The semantic validity of learned label prototypes is not analyzed (e.g., nearest neighbors, clustering), limiting interpretability claims.
3. The paper lacks details on parameter counts, FLOPs, memory/time costs, and scaling with views/labels, limiting reproducibility for large-scale settings.
4. The author has proposed a complex mutual information framework, but lacks a clear explanation of the motivation
5. How are missing views and labels considered? There is a lack of dedicated discussion on this key issue.

**Questions:**

Refer to Weaknesses

---

### Official Review · Reviewer_w46X · 2025-11-01

**Soundness:** 2
**Presentation:** 2
**Contribution:** 2
**Rating:** 2
**Confidence:** 5

**Summary:**

This paper introduces TITRL, a method for Incomplete Multi-View Multi-Label Classification (iMvMLC) designed to address concurrent missing views and labels. The method proposes a dual-layer constraint framework from an information-theoretic perspective, deriving variational bounds to guide the extraction of shared, task-relevant representations. TITRL employs a dual-fusion strategy, combining a Product-of-Experts-based early fusion for representation integration with a confidence-based late fusion for merging predictions. Additionally, it models sample-level label correlations using prototypes. The authors present experimental results on six datasets, comparing TITRL against nine baseline methods, and report superior performance.

**Strengths:**

The paper addresses the iMvMLC problem, which is a practical and challenging research direction. Furthermore, the authors provide a relatively thorough empirical evaluation across six datasets against nine baseline methods, reporting strong performance even under high data missingness rates.

**Weaknesses:**

1.The Sec 2.2 is very opaque, and the derivation suffers from logical leaps and ambiguities. For example, the derivation for the information decomposition in Eq.3 is unconvincing, as this form is not a standard decomposition of conditional mutual information and requires rigorous proof. Furthermore, the derivation for Eq.10 (detailed in Appendix A.2, Eq.30-32) is questionable. Transforming the minimization of $H(y|z^{(v)}) - H(y|x^{(v)})$ directly into a KL-divergence term is not valid as their supports differ. Even if this were accepted as a surrogate objective, the objective should be $D_{KL}(p(y|x^{(v)})||p(y|z^{(v)}))$ rather than the $D_{KL}(p(y|z^{(v)})||p(y|x^{(v)}))$ as claimed in Eq.10.

2.The paper omits critical implementation details, hindering reproducibility. For instance, the notations $p(y|x^{(v)})$, $p(y|z^{(v)})$, $p(y|\{ z \})$, and $p(y|\{z^{\sim(v)}\})$ are used extensively throughout Sec 2.2 and in the loss functions without a clear definition. It is unclear if these are generated by extra categorical prediction heads, whether such heads are shared or separate, or if they represent distribution parameters or concrete predictions.

3.The ablation study in Table 2 is insufficient to validate the specific contributions of the model's key components. It only removes broad loss terms ($\mathcal{L}_f, \mathcal{L}_c, \mathcal{L}_a$) or a part of the fusion mechanism. Given that TITRL’s framework is very similar to prior work like SIP (Liu et al., 2024), the primary additions appear to be the $\mathcal{L}_c$ loss and the confidence-based fusion mechanism. A more targeted ablation is required to isolate the impact of these specific additions. For example, the study does not show the performance when only the late fusion is removed, or when only the $\mathcal{L}_c$ loss is removed.

**Questions:**

In practice, how is $p(y|\{z^{\sim(v)}\})$ computed? Does this require $V$ separate leave-one-out fusion and prediction steps per iteration? If so, what is the computational overhead compared to the baselines?

---

### Note · Authors · 2025-11-12

I have read and agree with the venue's withdrawal policy on behalf of myself and my co-authors.